# Molecular features underlying differential SHP1/SHP2 binding of immune checkpoint receptors

Xiaozheng Xu[1†], Takeya Masubuchi[1†], Qixu Cai[2], Yunlong Zhao[1], Enfu Hui[1*]

[1]Section of Cell & Developmental Biology, Division of Biological Sciences, University of California, San Diego, La Jolla, United States; [2]Division of Life Science, State Key Laboratory of Molecular Neuroscience, Hong Kong University of Science and Technology, Hong Kong, China

**Abstract** A large number of inhibitory receptors recruit SHP1 and/or SHP2, tandem-SH2-containing phosphatases through phosphotyrosine-based motifs immunoreceptor tyrosine-based inhibitory motif (ITIM) and immunoreceptor tyrosine-based switch motif (ITSM). Despite the similarity, these receptors exhibit differential effector binding specificities, as exemplified by the immune checkpoint receptors PD-1 and BTLA, which preferentially recruit SHP2 and SHP1, respectively. The molecular basis by which structurally similar receptors discriminate SHP1 and SHP2 is unclear. Here, we provide evidence that human PD-1 and BTLA optimally bind to SHP1 and SHP2 via a bivalent, parallel mode that involves both SH2 domains of SHP1 or SHP2. PD-1 mainly uses its ITSM to prefer SHP2 over SHP1 via their C-terminal SH2 domains (cSH2): swapping SHP1-cSH2 with SHP2-cSH2 enabled PD-1:SHP1 association in T cells. In contrast, BTLA primarily utilizes its ITIM to prefer SHP1 over SHP2 via their N-terminal SH2 domains (nSH2). The ITIM of PD-1, however, appeared to be de-emphasized due to a glycine at pY+1 position. Substitution of this glycine with alanine, a residue conserved in BTLA and several SHP1-recruiting receptors, was sufficient to induce PD-1:SHP1 interaction in T cells. Finally, structural simulation and mutagenesis screening showed that SHP1 recruitment activity exhibits a bell-shaped dependence on the molecular volume of the pY+1 residue of ITIM. Collectively, we provide a molecular interpretation of the SHP1/SHP2-binding specificities of PD-1 and BTLA, with implications for the mechanisms of a large family of therapeutically relevant receptors.

## Editor's evaluation

This study elegantly addressed the SHP1/SHP2 preferences of ITIM/ITSM-containing inhibitory immunoreceptors PD-1 and BTLA, with solid evidence from cell-based, biochemical, biophysical, and domain-swapping assays. Importantly, it lays the foundation for further structural, physiological, and therapeutic studies.

**\*For correspondence:**
enfuhui@ucsd.edu

[†]These authors contributed equally to this work

**Competing interest:** The authors declare that no competing interests exist.

## Introduction

A wide spectrum of biological functions, including cell growth, survival, proliferation, differentiation, adhesion, migration, and communication, critically depend on tyrosine phosphorylations that occur on both cell surface receptors and intracellular effectors. Phosphotyrosines (pY) interact specifically with Src-homology-2 (SH2) domains to regulate protein-protein interactions and protein conformations (*Sadowski et al., 1986*; *Waksman et al., 1992*).

Two sets of enzymes reciprocally control tyrosine phosphorylation: protein tyrosine kinases (PTKs) that catalyze tyrosine phosphorylation and protein tyrosine phosphatases (PTPases) that catalyze the removal of phosphates from pY residues (*Denu and Dixon, 1998*; *Paul and Lombroso, 2003*; *Senis, 2013*; *Tonks, 2006*). Whereas a subset of PTKs and PTPases are anchored to cell membranes (e.g., receptor tyrosine kinases and receptor-like PTPases), others are cytoplasmic that are recruited by membrane receptors in response to an environmental cue.

SHP1 (PTPN6) and its paralog SHP2 (PTPN11) are cytoplasmic PTPases that are crucial for a wide range of cellular functions. Their dysregulation, due to either mutations or aberrant expression, contributes to a number of human pathologies, particularly cancer (*Bard-Chapeau et al., 2011*). SHP099, an allosteric inhibitor of SHP2 (*Chen et al., 2016*), is being evaluated in multiple clinical trials for cancer. SHP1 and SHP2 are structurally similar, both contain tandem-SH2 domains followed by a catalytic domain, and are coexpressed in multiple cell types. However, they are not redundant and contribute to different aspects of cellular functions (*Lorenz, 2009*; *Poole and Jones, 2005*). The biochemical basis for these differences is unclear.

Among the many reported functions of SHP1 and SHP2, they are known as key effectors for numerous inhibitory immunoreceptors, which recruit SHP1 and/or SHP2 to repress phosphorylation-dependent stimulatory signaling. These receptors include PD-1, BTLA, and LAIR, which repress the functions of T and B lymphocytes; SIRPα, which inhibits phagocytosis of myeloid cells; KIR/Ly49, which prevent NK cells from killing self-cells; PECAM1 and G6b-B, which inhibit platelet functions (*Coxon et al., 2017*); as well as several members of Siglecs, Sialic-acid-recognizing receptors. Collectively, these receptors operate as 'immune checkpoints' essential for self-tolerance, but can also be subverted by cancers and viruses to escape immune destruction. PD-1 blockade antibodies have produced impressive clinical activity against a subset of human cancer. There is also substantial interest in targeting other inhibitory receptors, or SHP1/SHP2 to overcome resistance to PD-1 targeted therapy (*Chen et al., 2020*), with promising results in mouse tumor models.

Common to SHP1/SHP2-recruiting immunoreceptors is the presence of one or both types of pY-based motifs in their intracellular domains (ICD): immunoreceptor tyrosine-based inhibitory motif (ITIM, consensus sequence S/I/V/LxYxxI/V/L) (*Burshtyn et al., 1996*; *Daëron et al., 1995*) and immunoreceptor tyrosine-based switch motif (ITSM, consensus sequence TxYxxV/I) (*Cannons et al., 2011*). Once phosphorylated, ITIM and ITSM act as docking sites for the SH2 domains of SHP1/SHP2. Moreover, ITSM in some receptors interacts with SH2-containing adaptor proteins SH2D1A and SH2D1B (*Cannons et al., 2011*).

Despite the general presumption that ITIM/ITSM-containing receptors recruit both SHP1 and SHP2, increasing evidence suggests that these receptors exhibit differential phosphatase-binding specificities. For example, PD-1 strongly recruits SHP2, but not SHP1, in both T cells and B cells (*Okazaki et al., 2001*; *Yokosuka et al., 2012*). In contrast, BTLA prefers to recruit SHP1 over SHP2 (*Celis-Gutierrez et al., 2019*; *Mintz et al., 2019*; *Xu et al., 2020*). These binding preferences were consistent with the functional analyses: deletion of SHP2, but not of SHP1, markedly decreases the inhibitory function of PD-1 in T cells (*Xu et al., 2020*). In contrast, the inhibitory function of BTLA is severely reduced by SHP1 deletion but not SHP2 deletion from T cells (*Xu et al., 2020*). Notably, FcγIIRB, another ITIM-containing receptor, recruits neither SHP1 nor SHP2, but recruits and signals through the lipid phosphatase SHIP (*Ono et al., 1997*).

The distinct phosphatase preferences of PD-1, BTLA, and FcγIIRB are striking and puzzling given the similarities in their pY motifs and in the structures of SHP1/SHP2. This knowledge gap has made it difficult to predict the functions, redundancy, competition, or synergy of the many ITIM-bearing receptors. Addressing these questions has met several challenges. First, both SHP1 and SHP2 can potentially interact with dual phosphorylated receptors in multiple possible modes: monovalent, bivalent parallel, or bivalent antiparallel, etc. Second, there is no reported structure for dual phosphorylated PD-1 or BTLA interacting with SHP1 or SHP2. Third, binding assays using SH2 domains of SHP1/SHP2, as extensively used in previous studies, likely do not reflect the behaviors of full-length proteins in cells. This is because SHP1 and SHP2 undergo complex regulation due to intramolecular contacts (*Hof et al., 1998*; *Pádua et al., 2018*; *Yang et al., 2003*) and reportedly dephosphorylate their docking sites within the receptor (*Goyette et al., 2017*; *Hui et al., 2017*; *Yokosuka et al., 2012*).

In this study, we dissected the molecular mechanisms by which PD-1 and BTLA discriminate between SHP1 and SHP2. We measured the affinities of all potential pY:SH2 interactions involved in

PD-1 and BTLA recruitment of SHP1 and SHP2 using surface plasmon resonance (SPR) and identified the optimal binding orientations of both SHP1 and SHP2. We then measured the recruitment of full-length SHP1 and SHP2 to PD-1 microclusters in intact, stimulated T cells expressing similar amounts of either wild-type (WT) or domain-swapped mutant of PD-1. This clean, 'in-cell' recruitment assay enabled us to quantitatively measure the net recruitment of SHP1/SHP2 after integrating the various regulatory mechanisms (autoinhibition, autodephosphorylation, etc.), as well as biophysical parameters (avidity, stoichiometry, compartmentalization, etc.). Through these experiments, we identified differing features between SHP1 and SHP2, and between PD-1 and BTLA, that led to the specificity dichotomy. Specifically, we isolated a single residue in ITIM that gates the SHP1-binding activity. Our work sheds light on the effector-binding specificities of a growing list of immune checkpoint receptors.

## Results

### PD-1 recruits SHP2, but not SHP1, whereas BTLA prefers to recruit SHP1

Previous studies suggest that tyrosine-phosphorylated PD-1 recruits SHP2, but not SHP1, to suppress T cell activation (*Xu et al., 2020*; *Yokosuka et al., 2012*). To begin investigating the molecular basis of this specificity, we utilized an antigen-presenting cell (APC) – T cell coculture assay incorporating the PD-L1:PD-1 pathway. In this assay, PD-1-mGFP-transduced Jurkat T cells were stimulated with PD-L1-transduced Raji B cells (APCs) pulsed with superantigen staphylococcal enterotoxin E (SEE). After lysing the cell conjugates at desired time points, we immunoprecipitated (IP) PD-1-mGFP from the cell lysates and probed pY and co-precipitated SHP1 or SHP2 using immunoblots (IB). PD-1 became tyrosine phosphorylated and recruited SHP2, but not SHP1, in a time-dependent fashion (*Figure 1A*), consistent with prior studies (*Xu et al., 2020*; *Yokosuka et al., 2012*). By contrast, in a parallel coculture system containing HVEM-transduced

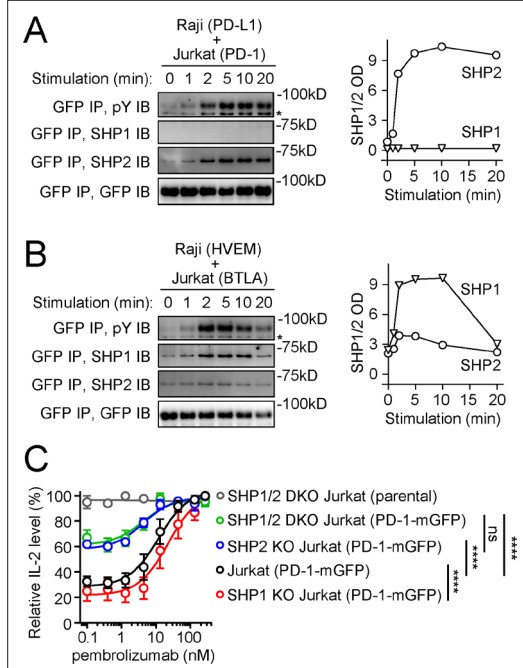

**Figure 1.** PD-1 recruits and signals through SHP2, but not SHP1, whereas BTLA prefers to recruit SHP1. (**A, B**) Left are representative immunoblots (IBs) showing the levels of bound SHP1 and SHP2 in PD-1-mGFP (**A**) or BTLA-mGFP (**B**) pulled down by GFP IP from indicated cell lysates, with the duration of stimulation prior to lysis indicated (see Materials and methods). IBs of GFP and phosphotyrosines (pY) of the same samples were shown to indicate PD-1 or BTLA input and their degrees of phosphorylation. Right are quantification graphs. (**C**) Relative IL-2 levels produced by PD-1-mGFP-expressing WT, SHP1 KO, SHP2 KO, or SHP1/2 DKO Jurkat cells stimulated with PD-L1-mCherry-expressing Raji cells in the presence of increasing concentrations of pembrolizumab (0, 0.4, 1.3, 4.4, 13.3, 44, 133, or 267 nM). For each type of Jurkat cells, IL-2 data were normalized to the condition with the highest IL-2 value in each replicate. Error bars are s.d. from three independent coculture assays run in three different days, with each assay run in technical triplicates. ****p<0.0001; ns, not significant; two-way ANOVA test.

The online version of this article includes the following source data and figure supplement(s) for figure 1:

**Source data 1.** Uncropped IBs for data shown in *Figure 1A and B*.

**Source data 2.** Raw data for *Figure 1* graphs.

**Figure supplement 1.** Flow cytometry histograms.

Raji cells and BTLA-mGFP-transduced Jurkat cells, BTLA recruited both SHP1 and SHP2, with a clear preference for SHP1 (*Figure 1B*), consistent with recent studies (*Celis-Gutierrez et al., 2019*; *Mintz et al., 2019*; *Xu et al., 2020*).

## SHP2 deletion, but not SHP1 deletion, decreased the magnitude of anti-PD-1 effect on IL-2 release

The little to no recruitment of SHP1 to PD-1 suggested that SHP1 minimally contributes to PD-1 function. To test this, we determined how deletion of SHP1, SHP2, or both from Jurkat cells affects PD-1 inhibition on IL-2 secretion. To measure the magnitude of PD-1 inhibitory effect, we used pembrolizumab (anti-PD-1) to precisely titrate PD-L1:PD-1 signaling. We created WT, SHP1 KO, SHP2 KO, and SHP1/SHP2 double KO (SHP1/2 DKO) Jurkat cells expressing similar levels of PD-1 (*Figure 1—figure supplement 1*) and stimulated these cells with SEE-pulsed Raji (PD-L1) cells at increasing concentrations of pembrolizumab. As expected, pembrolizumab dose-dependently increased IL-2 production from PD-1+ WT Jurkat cells (*Figure 1C*, black). Significantly less pembrolizumab-mediated IL-2 increase was observed in SHP2 KO cells, but not in SHP1 KO cells (*Figure 1C*, blue and red), arguing that SHP2, but not SHP1, contributes significantly to PD-1 function. Consistent with this notion, the magnitude of pembrolizumab effects was statistically indistinguishable in SHP1/2 DKO cells and in SHP2 KO cells (*Figure 1C*, green and blue).

## Both ITIM and ITSM contribute to the ability of PD-1 to recruit SHP2

We next attempted to clarify the relative contributions of ITIM and ITSM in mediating SHP2 recruitment by examining how mutations of these motifs affect SHP2 binding and PD-1 function. We generated Jurkat cells expressing similar levels of mGFP-tagged PD-1<sup>WT</sup>, PD-1<sup>FY</sup> (ITIM Y223 was mutated to phenylalanine), PD-1<sup>YF</sup> (ITSM Y248 was mutated to phenylalanine), or PD-1<sup>FF</sup> (both tyrosines were mutated) (*Figure 2—figure supplement 1*). Upon stimulation with PD-L1-transduced Raji cells, we detected SHP2, but not SHP1, in the PD-1<sup>WT</sup> IP (GFP IP), and as expected, SHP2 was undetectable in

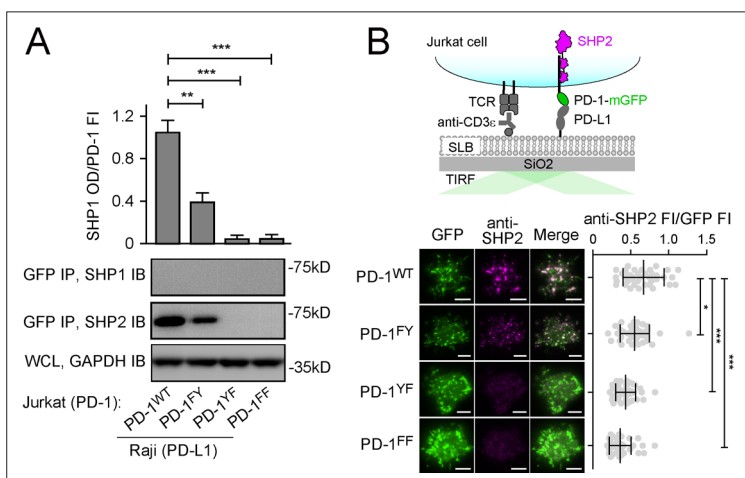

**Figure 2.** Both immunoreceptor tyrosine-based inhibitory motif (ITIM) and immunoreceptor tyrosine-based switch motif (ITSM) contribute to the ability of PD-1 to recruit SHP2. (**A**) Representative immunoblots (IBs) showing the levels of SHP1 and SHP2 bound to mGFP-tagged PD-1 variants pulled down from the indicated coculture lysates via GFP IP. GAPDH IB of the whole cell lysates (WCL) served as a loading control (see Materials and methods). Bar graphs on top summarize SHP2 optical density (OD) normalized to the fluorescence intensity (FI) of each PD-1 variant, based on flow cytometry data in *Figure 2—figure supplement 1*. Error bars are s.d. from three independent coculture experiments performed on different days. (**B**) Upper: a cartoon depicting a PD-1-mGFP-expressing Jurkat cell in contact with a supported lipid bilayer (SLB) containing anti-CD3 ε and PD-L1<sup>ECD</sup>. Lower left: representative TIRF images of both PD-1 (GFP) and endogenous SHP2 (stained with anti-SHP2) in an SLB-associated Jurkat expressing indicated PD-1 variants. Lower right: dot plots summarizing anti-SHP2 FI normalized to GFP FI of 40 Jurkat cells under each condition recorded on the same day with the same microscope setting (see Materials and methods); Error bars: s.d. Scale bars: 5 µm. *p<0.05; **p<0.01; ***p< 0.001; ns, not significant; Student's t-test.

The online version of this article includes the following source data and figure supplement(s) for figure 2:

**Source data 1.** Uncropped IBs for data shown in *Figure 2A*.

**Source data 2.** Raw data for quantification graphs in *Figure 2*.

**Figure supplement 1.** Flow cytometry histograms.

PD-1$^{FF}$ IP samples. The ITSM mutant PD-1$^{YF}$ also failed to recruit SHP2, whereas the ITIM mutant PD-1$^{FY}$ recruited SHP2, but significantly less than PD-1$^{WT}$ (*Figure 2A*). We confirmed these observations by visualizing PD-1:SHP2 interaction in intact T cells. We plated the foregoing Jurkat cells on a supported lipid bilayer (SLB) containing anti-CD3 $\varepsilon$ (for T cell receptor [TCR] stimulation) and recombinant PD-L1 ectodomain (PD-L1$^{ECD}$, for PD-1 stimulation). Total internal reflection microscopy (TIRF-M) in the GFP channel revealed PD-1 microclusters in all four cell types (*Figure 2B*). Immunostaining of SHP2 showed strong enrichment of SHP2 to PD-1$^{WT}$ microclusters. SHP2 recruitment was slightly weaker for PD-1$^{FY}$, but statistically significant (p=0.0306). SHP2 recruitment was almost completely abrogated in PD-1$^{YF}$, similar to the negative control PD-1$^{FF}$ (*Figure 2B*). These data are in general agreement with previous reports that ITSM is the dominant docking site for SHP2 (*Chemnitz et al., 2004*; *Okazaki et al., 2001*; *Patsoukis et al., 2020*; *Peled et al., 2018*; *Yokosuka et al., 2012*). However, our result showed that PD-1$^{WT}$ recruited more SHP2 than did PD-1$^{FY}$, suggesting that optimal SHP2 recruitment does require ITIM. The more obvious defect of PD-1$^{FY}$ in the co-IP assays might be due to the disruption of weak interactions by the non-equilibrium wash steps.

## PD-1-ITSM strongly prefers SHP2-cSH2 over SHP1-cSH2

We next investigated the molecular mechanism by which PD-1 recruits SHP2, but not SHP1, in T cells. SHP1 and SHP2 both contain two SH2 domains in tandem, the N-terminal SH2 (nSH2) and the C-terminal SH2 (cSH2). Our co-IP data indicated that SHP2 interacts with PD-1 in a bivalent fashion involving both SH2 domains. The bivalent interaction can potentially occur either in a parallel fashion in which nSH2 binds to ITIM and cSH2 binds to ITSM, or in an antiparallel fashion in which nSH2 and cSH2 bind to ITSM and ITIM, respectively. To determine the most favorable binding orientation, we

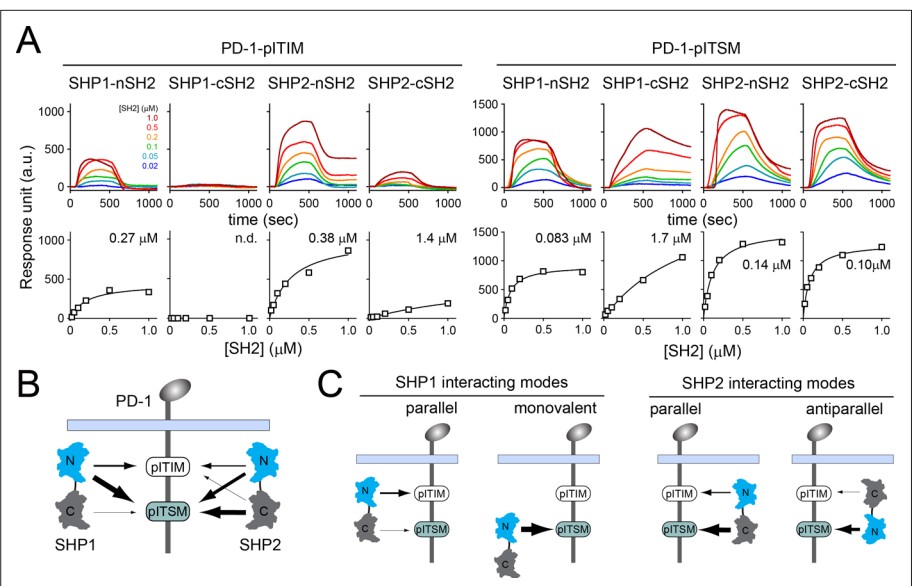

**Figure 3.** Surface plasmon resonance (SPR) measurements of binding between individual SH2 of SHP1 or SHP2 and immunoreceptor tyrosine-based inhibitory motif (ITIM) or immunoreceptor tyrosine-based switch motif (ITSM) of PD-1. (**A**) SPR sensorgrams (top) and the derived equilibrium-binding curves (bottom) showing the interactions of indicated SH2 and phosphorylated PD-1-ITIM (PD-1$^{YF}$) or PD-1-ITSM (PD-1$^{FY}$) immobilized onto Ni sensor chips. Individual SH2 proteins were injected at 20, 50, 100, 200, 500, and 1000 nM. Shown are representative of three independent experiments performed on three different sensorchips on three different days. The calculated $K_d$ values are indicated in the binding curves. (**B**) A cartoon depicting relative binding affinities of SHP1/SHP2 individual SH2 to PD-1-ITIM/ITSM, with the thickness of arrows matching the relative affinities calculated from the SPR data in (A). (**C**) Possible interacting modes of SHP1/SHP2-tSH2 with PD-1.

The online version of this article includes the following source data and figure supplement(s) for figure 3:

**Source data 1.** Uncropped SDS-PAGE gel images for *Figure 3—figure supplement 1A*.

**Source data 2.** Raw data for *Figure 3* and *Figure 3—figure supplement 1*.

**Figure supplement 1.** Single-molecule imaging monitoring SHP1/SHP2-tSH2 binding to PD-1 variants.

next measured the affinities for all the possible pY:SH2 interactions implicated in PD-1:SHP1 and PD-1:SHP2 interactions. We purified pre-phosphorylated ICDs of PD-1 mutants that contained only one tyrosine within either ITIM (PD-1$^{YF}$) or ITSM (PD-1$^{FY}$). We then used SPR to measure their binding affinities to purified SHP1-nSH2, SHP1-cSH2, SHP2-nSH2, and SHP2-cSH2 (*Figure 3A*) and summarized the dissociation constants ($K_d$) in *Supplementary file 1*. We also diagramed all the detectable pY:SH2 interactions in the context of tandem SH2 and PD-1$^{WT}$, with relative affinities depicted by arrow thickness. PD-1-ITSM is a better docking site than is PD-1-ITIM for each of the four SH2 tested (*Figure 3B*). A careful inspection of the data also revealed specific information, as detailed below.

For PD-1:SHP2 interactions, SHP2-nSH2 weakly preferred PD-1-ITSM ($K_d$ = 0.14 µM) over PD-1-ITIM ($K_d$ = 0.38 µM), and SHP2-cSH2 strongly preferred PD-1-ITSM ($K_d$ = 0.10 µM) over PD-1-ITIM ($K_d$ = 1.4 µM) (*Figure 3A and B*, *Supplementary file 1*). Thus, the parallel mode PD-1:SHP2 complex would be more energetically favorable than the antiparallel mode (*Figure 3C*, right, *Supplementary files 2 and 3*), consistent with a recent report (*Marasco et al., 2020*).

For PD-1:SHP1 interactions, SHP1-nSH2 preferred PD-1-ITSM ($K_d$ = 0.083 µM) over PD-1-ITIM ($K_d$ = 0.27 µM). Interestingly, SHP1-cSH2 appeared to be defective in PD-1 binding, exhibiting a rather weak affinity to PD-1-ITSM ($K_d$ = 1.7 µM), and no detectable binding to PD-1-ITIM (*Figure 3A and B*, *Supplementary file 1*). The inability of SHP1-cSH2 to bind PD-1-ITIM ruled out the antiparallel mode of PD-1:SHP1 interactions, but indicated the possibility of a monovalent mode in which SHP1-nSH2 interacts with PD-1-ITSM (*Figure 3C*, left). However, free energy calculations suggested that the parallel mode, which involves two SH2, is energetically favorable over the monovalent mode (*Supplementary files 2 and 3*).

To further examine the dominant mode of PD-1:SHP1 interactions, we employed a single-molecule assay to determine whether PD-1:SHP1 interactions require only ITSM, as would be expected for the monovalent mode, or both ITIM and ITSM, as would be expected for the bivalent parallel mode. We sparsely attached monomeric, fluorescently labeled, pre-phosphorylated and biotinylated PD-1$^{WT}$, PD-1$^{FY}$, or PD-1$^{YF}$ (*Figure 3—figure supplement 1A and B*) to a biotin polyethylene glycol (PEG)-coated coverslip via streptavidin. TIRF-M resolved individual PD-1 monomers as discrete spots, which underwent photobleaching in single steps (*Figure 3—figure supplement 1C and D*). After the addition of JF646-labeled tSH2 of either SHP1 or SHP2, we visualized PD-1:tSH2 interaction at the coverslip (*Figure 3—figure supplement 1E*, left). Recruitment of SHP2-tSH2 to PD-1 led to the appearance of JF646 signal that colocalized with PD-1 molecules (*Figure 3—figure supplement 1E*, upper right). Each SHP2-tSH2 spot typically persisted for several seconds, then disappeared due to dissociation from PD-1, leading to a step-like time course (*Figure 3—figure supplement 1E*, lower right). As expected, SHP1-tSH2 displayed a lower degree of PD-1 occupancy (*Figure 3—figure supplement 1F*). Moreover, mutation of either ITIM (PD-1$^{FY}$) or ITSM (PD-1$^{YF}$) strongly reduced the PD-1 occupancy for both SHP1-tSH2 and SHP2-tSH2 as compared to the WT control (PD-1$^{WT}$) (*Figure 3—figure supplement 1F*), further supporting that both SHP1-tSH2 and SHP2-tSH2 optimally bind to PD-1$^{WT}$ in a bivalent fashion involving both ITIM and ITSM.

Collectively, data presented in this section demonstrated that both SHP1 and SHP2 interact with PD-1 primarily via the bivalent parallel orientation. However, the PD-1:SHP1 interaction is much less stable due to the very weak affinity between SHP1-cSH2 and PD-1-ITSM.

## Replacement of SHP1-cSH2 by SHP2-cSH2 is sufficient to induce PD-1:SHP1 association in T cells

The SPR data (*Figure 3B*, *Supplementary file 1*) indicated that the SHP1-cSH2 barely interacts with PD-1-ITSM, whereas SHP2-cSH2 displayed a 17-fold higher affinity to the ITSM of PD-1. Indeed, according to the NMR structure of PD-1-pITSM:SHP2-cSH2 complex (PDB code: 6R5G) (*Marasco et al., 2020*), multiple residues in SHP2-cSH2 (e.g., K120, M171, T205, T208) that contribute to the interaction with PD-1-ITSM are replaced in SHP1-cSH2 (*Figure 4A*). Thus, we next determined if swapping the cSH2 of SHP1 with that of SHP2 could induce PD-1:SHP1 binding in T cells. We sought to image the recruitment of domain-swapped chimeric mutants of SHP1 to PD-1 microclusters. To avoid competition from endogenous SHP1 and SHP2, we generated SHP1/2 DKO Jurkat cells and co-transduced PD-1-mGFP with mCherry-tagged SHP1$^{WT}$ or SHP1 mutant with one or both of its SH2 domains replaced by those of SHP2 (*Figure 4B*, left). Having confirmed that these cells expressed similar levels of PD-1-mGFP and mCherry-tagged SHP1 variants (*Figure 4—figure supplement 1*), we

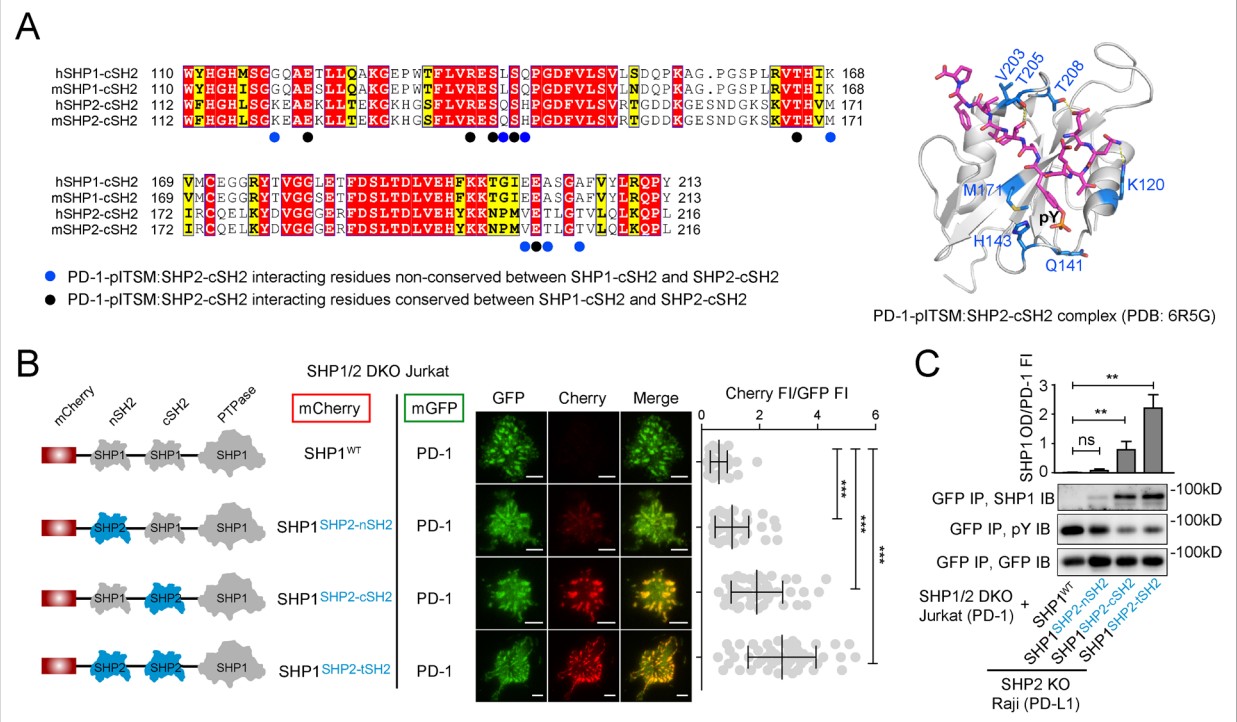

**Figure 4.** Swapping the cSH2 of SHP1 with that of SHP2 induced PD-1:SHP1 association in T cells. (**A**) Left: sequence alignment of cSH2 of human and mouse SHP1/SHP2. The underlying dots denote SHP2-cSH2 residues that participate in PD-1-pITSM binding, according to NMR structure of PD-1-pITSM:SHP2-cSH2 complex (PDB code: 6R5G): black dots highlight residues that are conserved in SHP1-cSH2; blue dots highlight residues that are not conserved in SHP1-cSH2. Right: NMR structure of PD-1-pITSM:SHP2-cSH2 complex (PDB code: 6R5G) with PD-1 depicted in a stick model and SHP2-cSH2 showed in a ribbon diagram, in which the blue-dot-denoted residues in the sequence alignment are highlighted in blue sticks. (**B**) Left: diagram showing the design of mCherry-tagged, SH2-swapped SHP1 variants, with one or both of its SH2 replaced with that of SHP2. Middle: representative TIRF images of PD-1 (GFP) and SHP1 variants (mCherry) in supported lipid bilayer (SLB)-associated SHP1/2 DKO Jurkat cells expressing PD-1-mGFP and mCherry-SHP1 variants. Right: dot plots summarizing mCherry fluorescence intensity (FI) normalized to GFP FI of 40 Jurkat cells under each condition recorded on the same day with the same microscope setting. Error bars: s.d. Scale bars: 5 µm. (**C**) Representative immunoblots (IBs) of mCherry-SHP1 variants co-precipitated with PD-1-mGFP from indicated cell lysates. IBs of GFP indicate PD-1 input. IBs of phosphotyrosines (pY) indicate PD-1 phosphorylation. Bar graphs summarize optical density (OD) of SHP1 variants normalized to the FI of PD-1, based on flow cytometry data in *Figure 4—figure supplement 1*. Error bars are s.d. from three independent coculture experiments performed on different days. *p<0.05; **p<0.01; ***p<0.001; ns, not significant; Student's t-test.

The online version of this article includes the following source data and figure supplement(s) for figure 4:

**Source data 1.** Uncropped IBs for data shown in *Figure 4C*.

**Source data 2.** Raw data for quantification graphs in *Figure 4*.

**Figure supplement 1.** Flow cytometry histograms.

stimulated each type of Jurkat cells with an SLB containing anti-CD3 $\varepsilon$ and PD-L1[ECD]. TIRF-M showed that upon cell-bilayer contact, PD-1 formed microclusters that recruited little to no SHP1[WT], as manifested by nearly undetectable mCherry signal in the GFP foci (*Figure 4B*, top row), consistent with previous reports (*Xu et al., 2020*; *Yokosuka et al., 2012*). Swapping the nSH2 of SHP1 with that of SHP2 (SHP1[SHP2-nSH2]) increased the mCherry signal in PD-1 microclusters, but only to a minor degree (*Figure 4B*, second row). In contrast, swapping the cSH2 of SHP1 with that of SHP2 (SHP1[SHP2-cSH2]) led to a marked increase in mCherry signal in the PD-1 microclusters (*Figure 4B*, third row), to a comparable extent as SHP1[SHP2-tSH2], in which both SH2 of SHP1 were replaced with those of SHP2 (*Figure 4B*, bottom row).

We confirmed the TIRF results using a co-IP assay. After stimulation of the foregoing Jurkat cells (*Figure 4B*) with PD-L1-transduced Raji cells, we pulled down PD-1-mGFP and examined the co-precipitated SHP1 variants using IB. We detected no signal of SHP1[WT], weak signal of SHP1[SHP2-nSH2], and strong signal of SHP1[SHP2-cSH2] and SHP1[SHP2-tSH2] (*Figure 4C*, GFP IP, SHP1 IB). Notably, PD-1 phosphorylation was inversely correlated with the recruitment of the SHP1 variants (*Figure 4C*, GFP IP, pY IB),

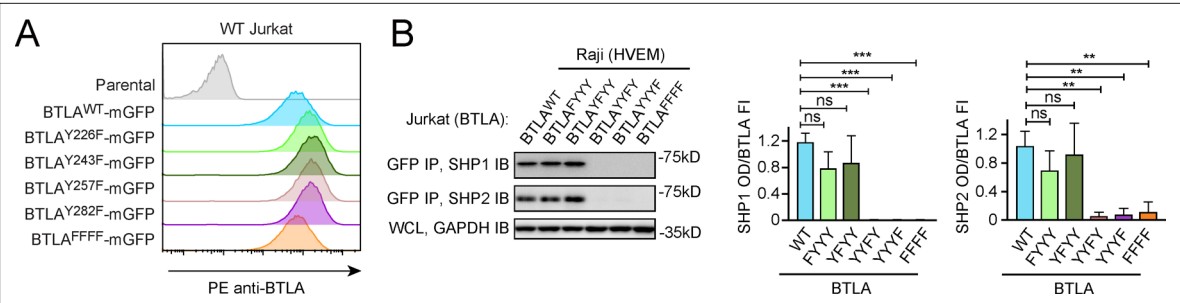

**Figure 5.** Both immunoreceptor tyrosine-based inhibitory motif (ITIM) and immunoreceptor tyrosine-based switch motif (ITSM) are required for BTLA to recruit SHP1/SHP2. (**A**) Flow cytometry histograms showing BTLA surface expressions in the indicated Jurkat cells. (**B**) Left: representative immunoblots (IBs) showing the levels of SHP1 and SHP2 bound to the mGFP-tagged BTLA variants captured by GFP IP. GAPDH IB of the whole cell lysates (WCL) served as a loading control. Right: bar graphs summarizing SHP1 optical density (OD) and SHP2 OD normalized to the fluorescence intensity (FI) of the corresponding BTLA variants, based on flow cytometry data in (**A**). Error bars are s.d. from three independent coculture experiments performed on different days. **p<0.01; ***p<0.001; ns, not significant; Student's t-test (n = 3).

The online version of this article includes the following source data and figure supplement(s) for figure 5:

Source data 1. Uncropped IBs for *Figure 5B* and *Figure 5—figure supplement 1B*.

Source data 2. Raw data for bar graphs in *Figure 5* and *Figure 5—figure supplement 1*.

Figure supplement 1. Y226 and Y243 are dispensable for SHP1/SHP2 recruitment by BTLA.

---

supporting the notion that PD-1 is a substrate for its bound PTPases (*Goyette et al., 2017*; *Hui et al., 2017*; *Yokosuka et al., 2012*). Collectively, data reported in this section demonstrated that cSH2 is the major determinant underlying PD-1's strong preference for SHP2 over SHP1.

## ITIM and ITSM are both required for BTLA to recruit SHP1/SHP2

Having established deficient ITSM:cSH2 interactions as the primary basis for the weak stability of PD-1:SHP1 binding, we next turned our attention to receptors that normally recruit SHP1 in T cells, such as BTLA (*Figure 1B*), to gain further insights into the mechanisms of effector PTPase discrimination by ITIM/ITSM-bearing receptors (*Gavrieli et al., 2003*; *Mintz et al., 2019*; *Xu et al., 2020*). We wished to determine the structural features in BTLA that enabled its SHP1 recruitment in T cells.

The ICD of human BTLA harbors four phosphorylatable tyrosines (Y226, Y243, Y257, and Y282), in which Y257 and Y282 are embedded in ITIM and ITSM, respectively. We first asked which tyrosine(s) of BTLA are required for SHP1 and SHP2 recruitment. Analogous to PD-1 assays (*Figure 2A*), we established Jurkat cell lines expressing either WT or mutant BTLA in which one of the four phosphorylatable tyrosines was replaced by phenylalanine (*Figure 5A*). We then stimulated these cell lines, in parallel, with HVEM-expressing Raji B cells. Co-IP experiments showed that mutation of either or both of the non-ITIM/ITSM phosphorylatable tyrosines (Y226F: BTLA$^{FYYY}$; Y243F: BTLA$^{YFYY}$; Y226F and Y243F: BTLA$^{FFYY}$) had little to no effect on the abilities of BTLA to recruit SHP1/SHP2 (*Figure 5*, *Figure 5—figure supplement 1*), demonstrating that these two tyrosines are dispensable for BTLA-mediated recruitment of SHP1/SHP2. In contrast, mutation of either ITIM tyrosine (Y257F: BTLA$^{YYFY}$) or ITSM tyrosine (Y282F: BTLA$^{YYYF}$) abolished the binding of both SHP1 and SHP2 (*Figure 5B*), consistent with previous studies (*Gavrieli et al., 2003*). Thus, ITIM and ITSM are both necessary for SHP1 and SHP2 recruitment by BTLA. These data also suggest that both SH2 domains of SHP1 and SHP2 are required for their recruitment to BTLA. The more stringent requirement of both ITIM and ITSM of BTLA indicates that its ITIM and ITSM play more balanced roles in mediating SHP1/SHP2 recruitment than those of PD-1.

## SHP1 and SHP2 both interact with BTLA via the bivalent parallel mode

We next sought to determine the most favorable binding orientations for BTLA:SHP1 interactions and BTLA:SHP2 interactions. Analogous to SPR assays with PD-1 (*Figure 3*), we measured the affinities of individual pY:SH2 interaction implicated in BTLA:SHP1/SHP2 interactions using sensor chips coated with pre-phosphorylated BTLA triple-tyrosine-mutant FFYF (BTLA-ITIM), which contained a lone tyrosine (Y257) in its ITIM, or pre-phosphorylated BTLA triple-tyrosine-mutant FFFY (BTLA-ITSM),

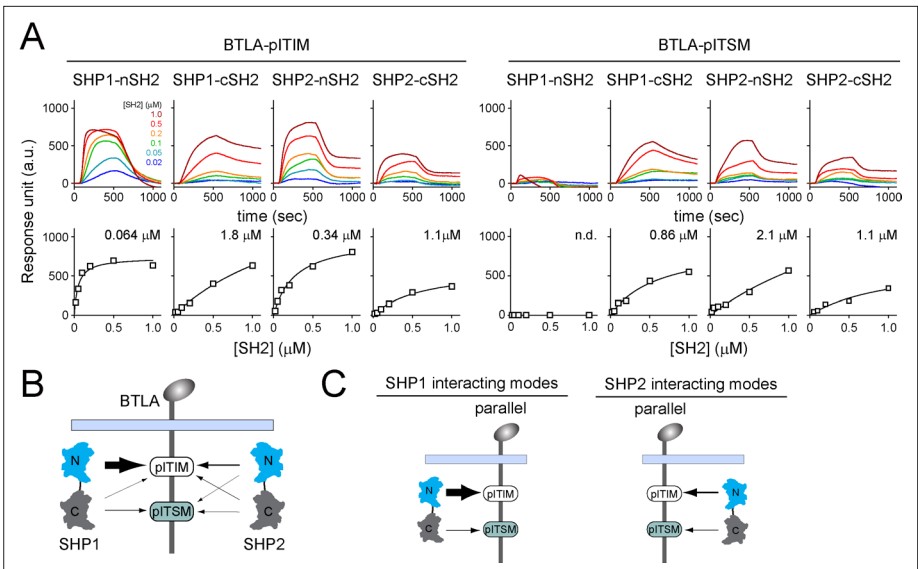

**Figure 6.** Surface plasmon resonance (SPR) measurements of binding between individual SH2 of SHP1 or SHP2 and immunoreceptor tyrosine-based inhibitory motif (ITIM) or immunoreceptor tyrosine-based switch motif (ITSM) of BTLA. (**A**) SPR sensorgrams (top) and the derived equilibrium binding curves (bottom) showing the interactions of indicated SH2 and phosphorylated BTLA-ITIM (BTLA FFYF) or BTLA-ITSM (BTLA FFFY) immobilized to Ni sensor chips. Individual SH2 proteins were injected at 20, 50, 100, 200, 500, and 1000 nM. Shown are representative of three independent experiments performed on three different sensorchips on three different days. The calculated $K_d$ values are indicated in the binding curves. (**B**) A cartoon depicting relative binding affinities of SHP1/SHP2 individual SH2 to BTLA-ITIM/ITSM, with the thickness of arrows matching the relative affinities calculated from the SPR data in (**A**). (**C**) Possible interacting modes of SHP1/SHP2-tSH2 with BTLA.

The online version of this article includes the following source data for figure 6:

**Source data 1.** Raw data for equilibrium binding curves in **Figure 6A**.

---

which contained a lone tyrosine (Y282) in its ITSM (**Figure 6A**). These experiments revealed that for both SHP1 and SHP2, their nSH2 and cSH2 domains prefer BTLA-ITIM and BTLA-ITSM, respectively (**Figure 6A and B**, **Supplementary file 1**). Thus, we concluded that the most favorable BTLA:SHP1 and BTLA:SHP2 interactions both occur in a parallel mode (**Figure 6C**), similar to PD-1:SHP1 and PD-1:SHP2 interactions.

## BTLA-ITIM is a high-affinity docking site for SHP1-nSH2

On a closer inspection of the SPR data, we found that the relative contribution of ITIM and ITSM in BTLA is opposite to that in PD-1. While the ITSM is the major SH2 docking site in PD-1, the ITIM appeared to be the major SH2 docking site in BTLA. This is particularly striking in the case of BTLA:SHP1 interaction: SHP1-nSH2 exhibited an impressive affinity to BTLA-ITIM ($K_d$ = 0.064 μM), 13-fold higher than the affinity between SHP1-cSH2 and BTLA-ITSM ($K_d$ = 0.86 μM) (**Figure 6A**, **Supplementary file 1**).

Notably, the affinity of SHP1-nSH2 to BTLA-ITIM ($K_d$ = 0.064 μM) was also four-fold higher than its affinity to PD-1-ITIM ($K_d$ = 0.27 μM) (**Figure 6A**, **Supplementary file 1**). Thus, even though BTLA-ITSM is a poor docking site for SHP1-cSH2, akin to PD-1-ITSM, BTLA-ITIM is a much better docking site for SHP1-nSH2 than is PD-1-ITIM. Conceivably, the strong BTLA-ITIM:SHP1-nSH2 interaction may compensate for the weak BTLA-ITSM:SHP1-cSH2 interaction, leading to an overall stable BTLA:SHP1 association in T cells.

## Swapping PD-1-ITIM with BTLA-ITIM induced PD-1:SHP1 interaction in T cells

The foregoing data support a hypothesis that the stability of ITIM:SH2 interactions is vital for ITIM/ITSM-bearing receptors to recruit SHP1. To test this experimentally, we assayed whether replacing the 'low-affinity' ITIM of PD-1 with the 'high-affinity' ITIM of BTLA could induce PD-1:SHP1 interaction. We

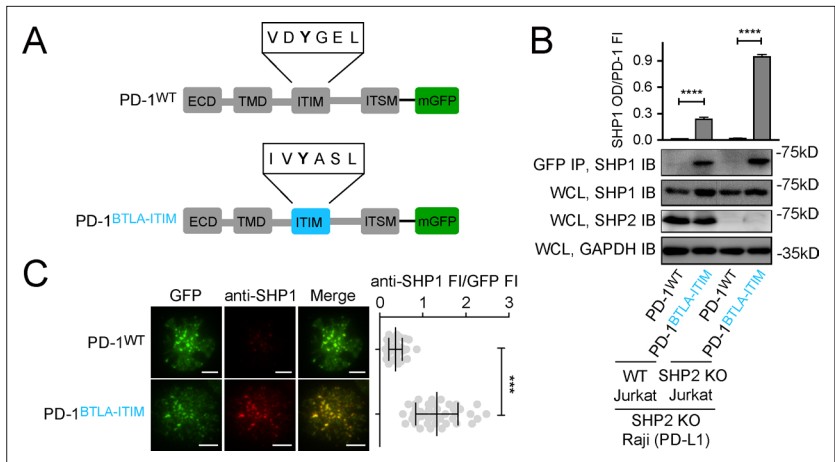

**Figure 7.** Swapping PD-1-ITIM with BTLA-ITIM induced PD-1:SHP1 interaction in T cells. (**A**) Cartoons depicting the domains and motifs of PD-1$^{WT}$-mGFP and PD-1$^{BTLA-ITIM}$-mGFP. (**B**) Representative immunoblots (IBs) showing the levels of SHP1 bound to mGFP-tagged PD-1 variants pulled down from the indicated coculture lysates via GFP IP. SHP1 IB and SHP2 IB of the whole cell lysates (WCL) indicate their inputs. GAPDH IB of the WCL served as a loading control. Bar graphs summarize SHP1 optical density (OD) normalized to the fluorescence intensity (FI) of each PD-1 variant, based on flow cytometry data in *Figure 7—figure supplement 1*. Error bars are s.d. from three independent coculture experiments performed on three different days. (**C**) Left: representative TIRF images of both PD-1 (GFP) and endogenous SHP1 (stained with anti-SHP1) in a supported lipid bilayer (SLB)-associated SHP2 KO Jurkat expressing indicated PD-1 variants. Right: dot plots summarizing anti-SHP1 FI normalized to GFP FI of 40 Jurkat cells under each condition recorded on the same day using the same microscope setting; Error bars: s.d. Scale bars: 5 µm. ***p<0.001; ****p<0.0001; Student's t-test.

The online version of this article includes the following source data and figure supplement(s) for figure 7:

**Source data 1.** Uncropped IBs for *Figure 7B* and *Figure 7—figure supplement 2B*.

**Source data 2.** Raw data for quantification graphs in *Figure 7* and *Figure 7—figure supplement 2*.

**Figure supplement 1.** Flow cytometry histograms.

**Figure supplement 2.** Replacing BTLA-ITIM with PD-1-ITIM abolished BTLA:SHP1 interaction in Jurkat cells.

transduced Jurkat cells with either mGFP-tagged PD-1$^{WT}$ or PD-1$^{BTLA-ITIM}$, in which we replaced PD-1-ITIM (VD**Y**GEL) with BTLA-ITIM (IV**Y**ASL) (*Figure 7A*, *Figure 7—figure supplement 1*). Following stimulation of both types of Jurkat cells using PD-L1-expressing Raji B cells, co-IP assays revealed that PD-1$^{BTLA-ITIM}$, but not PD-1$^{WT}$, recruited SHP1 (*Figure 7B*). We confirmed this finding using SHP2 KO Jurkat cells, which allowed us to examine PD-1:SHP1 interaction without potential competition from SHP2 (*Figure 7A and B*). We further verified these findings in intact cells using TIRF-M. In the cell-SLB assays, we observed microclusters of both PD-1$^{WT}$ and PD-1$^{BTLA-ITIM}$, and confirmed that the PD-1$^{BTLA-ITIM}$ microclusters recruited significantly more SHP1 than did the PD-1$^{WT}$ microclusters (*Figure 7C*). Finally, in a reciprocal set of experiments, we found that the replacement of the BTLA-ITIM with the PD-1-ITIM markedly decreased the SHP1 recruitment to BTLA in both WT Jurkat and SHP2 KO Jurkat cells (*Figure 7—figure supplement 2*). Together, data presented in this section demonstrated that SHP1 primarily discriminates BTLA from PD-1 based on their ITIMs.

## Replacement of the pY+1 glycine in PD-1-ITIM with alanine was sufficient to induce PD-1:SHP1 association in T cells

We noted that the BTLA-ITIM (IV**Y**ASL) differs from PD-1-ITIM (VD**Y**GEL) at four residues flanking pY: V221, D222, G224, and E225 in PD-1-ITIM are replaced by I, V, A, and S, respectively, in BTLA-ITIM (*Figure 8A*). We wished to determine which replacement contributed the most in inducing SHP1 binding of PD-1$^{BTLA-ITIM}$. We generated SHP2 KO Jurkat cells expressing comparable levels of PD-1$^{V221I}$, PD-1$^{D222V}$, PD-1$^{G224A}$, or PD-1$^{E225S}$, each containing a C-terminal mGFP (*Figure 8—figure supplement 1A*), and asked which mutants were able to recruit SHP1. We also used cells expressing PD-1$^{WT}$-mGFP and cells expressing PD-1$^{BTLA-ITIM}$-mGFP as controls. Upon stimulation with PD-L1-transduced Raji cells, we IP'ed PD-1-mGFP and blotted for SHP1. As expected, SHP1 signal was evident in the precipitates

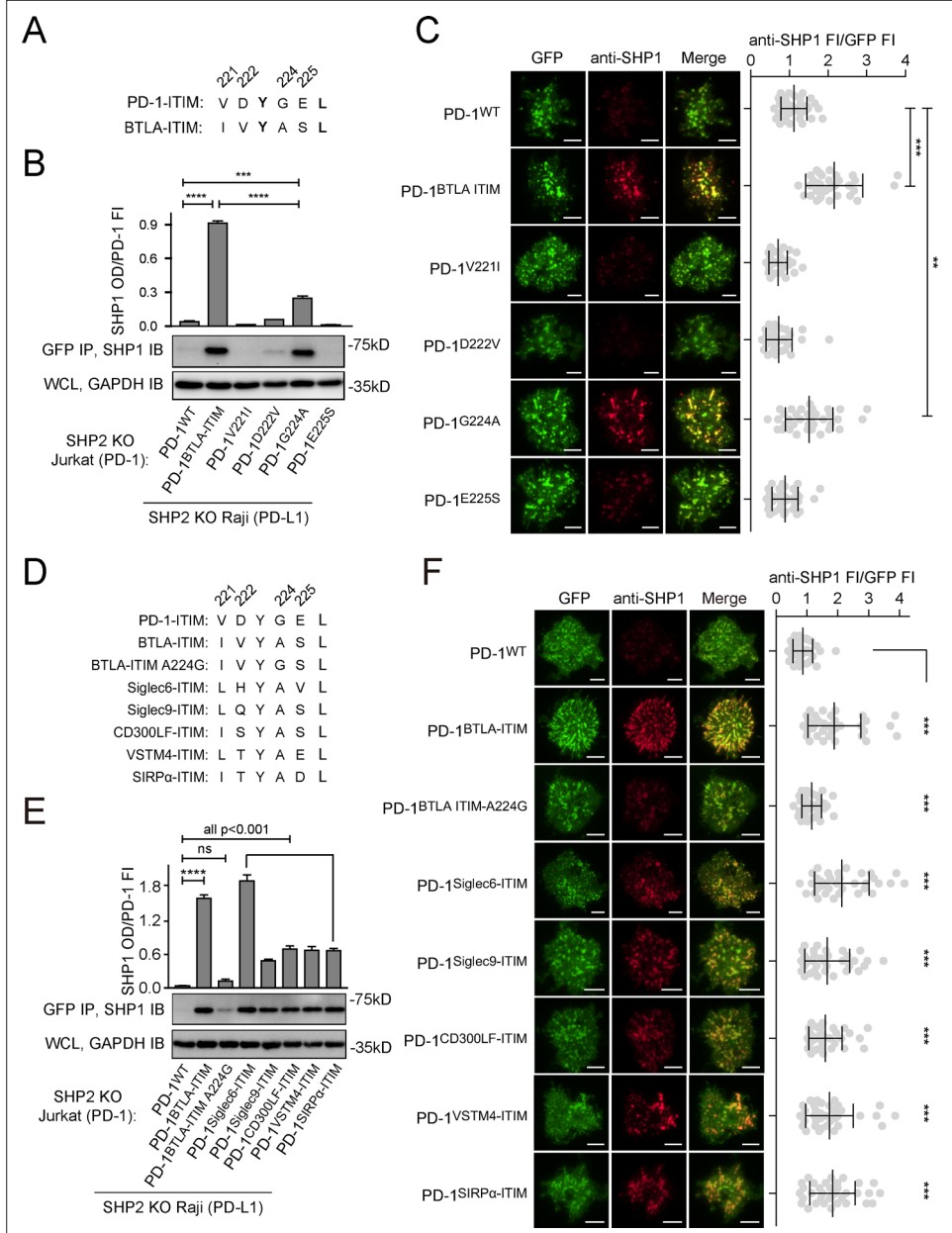

**Figure 8.** Glycine to alanine substitution at the pY+1 position of PD-1 immunoreceptor tyrosine-based inhibitory motif (ITIM) promoted SHP1 recruitment. (**A**) Cartoons depicting the amino acid (AA) alignment of PD-1-ITIM and BTLA-ITIM. (**B**) Representative immunoblots (IBs) showing the levels of SHP1 bound to mGFP-tagged PD-1 variants pulled down from the indicated coculture lysates via GFP IP. GAPDH IB of whole cell lysates (WCL) served as a loading control. Bar graphs summarize SHP1 optical density (OD) normalized to the fluorescence intensity (FI) of each PD-1 variant, based on flow cytometry data in *Figure 8—figure supplement 1A*. Error bars are s.d. from three independent coculture experiments conducted on three different days. (**C**) Left: representative TIRF images of both PD-1 (GFP) and endogenous SHP1 (stained with anti-SHP1) in a supported lipid bilayer (SLB)-associated SHP2 KO Jurkat cell expressing indicated PD-1 variants. Right: dot plots summarizing anti-SHP1 FI normalized to GFP FI of 35 Jurkat cells under each condition recorded on the same day using the same microscope setting (see Materials and methods). (**D**) Cartoons showing the AA sequences of ITIM of indicated receptors. (**E**) Representative IBs showing the levels of SHP1 co-precipitated with mGFP-tagged PD-1 variants, with the original ITIM replaced by the indicated ITIM, from the indicated coculture lysates via GFP IP. GAPDH IB of WCL served as a loading control. Bar graphs summarize SHP1 OD normalized to the FI of each PD-1 variant, based on flow cytometry data in *Figure 8—figure supplement 1C*. Error bars are s.d. from three independent coculture experiments conducted on three different days. (**F**) Left: representative TIRF images of both PD-1 (GFP) and endogenous SHP1 (stained

*Figure 8 continued on next page*

*Figure 8 continued*
with anti-SHP1) in an SLB-associated SHP2 KO Jurkat cell expressing PD-1 variants with indicated ITIM. Right: dot plots summarizing anti-SHP1 FI normalized to GFP FI of 35 Jurkat cells under each condition recorded on the same day using the same microscope setting (see Materials and methods); Error bars: s.d. Scale bars: 5 µm. \*\*p<0.01; \*\*\*p<0.001; \*\*\*\*p<0.0001; ns, not significant. Student's t-test.

The online version of this article includes the following source data and figure supplement(s) for figure 8:

**Source data 1.** Uncropped IBs for *Figure 8B and E*.

**Source data 2.** Raw data for quantification graphs in *Figure 8*.

**Figure supplement 1.** Flow cytometry histograms of PD-1 variants and sequence alignment of immunoreceptor tyrosine-based inhibitory motifs (ITIMs) in immune receptors.

---

of PD-1$^{BTLA-ITIM}$, but not in PD-1$^{WT}$. Notably, PD-1$^{G224A}$ recruited the most SHP1 among the single-point mutants, despite less than PD-1$^{BTLA-ITIM}$, the positive control (*Figure 8B*). These results suggest that the alanine residue at the pY+1 position of BTLA-ITIM plays a key role in SHP1 recruitment. However, the significantly lower SHP1 recruitment for PD-1$^{G224A}$ than for PD-1$^{BTLA-ITIM}$ (p=0.000016) indicates that other residues within the BTLA-ITIM also contribute to SHP1 binding to some extent, even though V221I, D222V, or E225S point mutation induced little PD-1:SHP1 association.

We validated the aforementioned findings using TIRF imaging of SLB-stimulated T cells. We observed PD-1 microclusters for all six PD-1 variants. Consistent with *Figure 7C*, microclusters of PD-1$^{BTLA-ITIM}$, but not PD-1$^{WT}$, recruited SHP1 (*Figure 8C*, rows 1 and 2). Among the single-point mutants, only PD-1$^{G224A}$ microclusters clearly recruited SHP1 (*Figure 8C*, row 5), albeit less than did PD-1$^{BTLA-ITIM}$ microclusters when SHP1 signal was normalized to the PD-1 signal. In contrast, the other three single-point mutants (PD-1$^{V221I}$, PD-1$^{D222V}$, and PD-1$^{E225S}$) showed little to no SHP1 recruitment (*Figure 8C*, rows 3, 4, and 6).

Indeed, sequence alignment revealed that alanine is conserved at pY+1 position of ITIM in several inhibitory receptors (*Figure 8—figure supplement 1B*), including Siglec6, Siglec9, CD300LF, VSTM4, and SIRPα, most of which reportedly recruit SHP1 (*Alvarez-Errico et al., 2004*; *Avril et al., 2004*; *Crocker et al., 2007*; *Sui et al., 2004*; *Veillette et al., 1998*). As expected, swapping the PD-1-ITIM by the ITIM of the foregoing inhibitory receptors (*Figure 8D*, *Figure 8—figure supplement 1C*) significantly increased SHP1 recruitment to PD-1 immunoprecipitates and PD-1 microclusters as compared to PD-1$^{WT}$ (*Figure 8E and F*).

## A medium-sized nonpolar residue at pY+1 position of the ITIM is required for SHP1 recruitment

The ability of the somewhat conservative mutation (G224A) to induce PD-1:SHP1 binding was unexpected; however, given that alanine has a larger side chain than does glycine, we next sought to determine how the side chain property of pY+1 position influences PD-1:SHP1 interaction. We first simulated the structure of PD-1-pITIM:SHP1-nSH2 through homology modeling based on the structure of PD-1-pITIM:SHP2-nSH2 (PDB code: 6ROY) (*Marasco et al., 2020*). The simulation revealed a hydrophobic pocket within SHP1-nSH2 that likely coordinates the side chain of pY+1 alanine (*Figure 9A*). Further structural simulation suggests that the size of the hydrophobic pocket is a good fit for a medium-sized nonpolar residue alanine, valine, leucine, or isoleucine, but not for a larger residue phenylalanine or tryptophan (*Figure 9B*). To test this experimentally, we mutated the glycine to a series of residues that differ in the size and polarity of their side chains. We established Jurkat lines that express each of the PD-1 mutants fused to a GFP tag at comparable levels indicated by flow cytometry (*Figure 9—figure supplement 1A*). In the cell-SLB assay, we found that a nonpolar residue with a medium-sized side-chain (G224A, G224V, G224L, G224I) but not a bulky or rigid side-chain (G224F, G224W, and G224P) at pY+1 position of PD-1-ITIM strongly promoted PD-1:SHP1 association (*Figure 9C*, rows 3–9). Indeed, plotting the SHP1 recruitment against the molecular volume of the seven nonpolar residues, excluding the rigid proline, revealed a bell-shaped dependence that peaked at leucine and isoleucine (*Figure 9D*). Finally, a polar or charged residue at this position of PD-1 failed to induce SHP1 recruitment, as observed for G224S, G224T, G224K, and G224D mutants (*Figure 9C*, rows 10–13). These results validated the foregoing structural modeling that a hydrophobic pocket in SHP1-nSH2 coordinates the side chain of the pY+1 residue of ITIM (*Figure 9B*). Finally, to determine

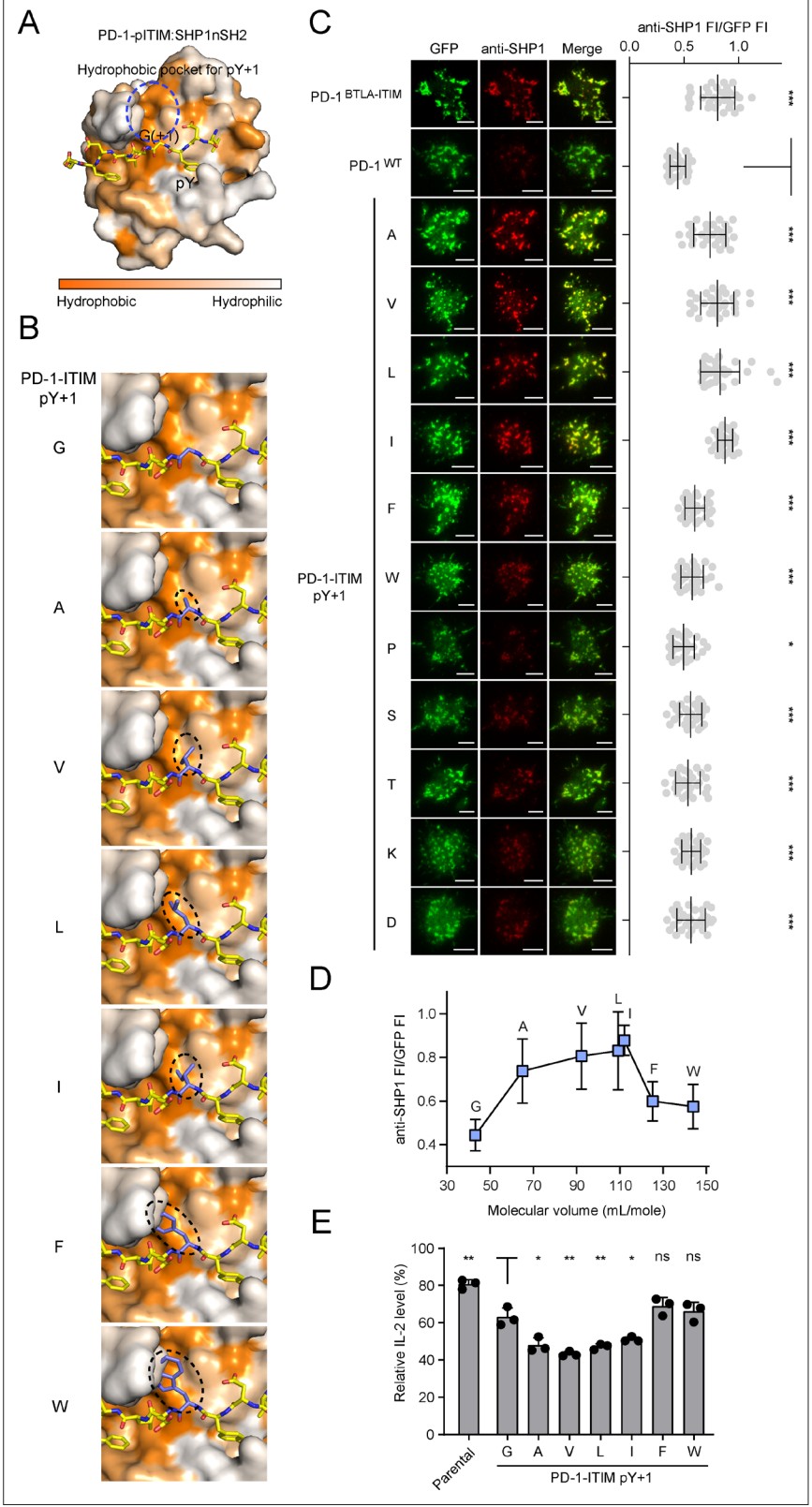

**Figure 9.** A medium-sized nonpolar residue at pY+1 position of the immunoreceptor tyrosine-based inhibitory motif (ITIM) is optimal for SHP1 recruitment. (**A**) The combined surface (SHP1-nSH2) and stick (PD-1-pITIM) representation showing the hydrophobic pocket (highlighted by a blue circle) in SHP1-nSH2 for coordinating pY+1 residue of PD-1-pITIM. This structural model of SHP1-nSH2:PD-1-pITIM complex was generated by homology

*Figure 9 continued on next page*

*Figure 9 continued*

modeling using the SWISS-Model based on the crystal structure of SHP2-nSH2:PD-1-pITIM complex (PDB: 6ROY). (**B**) SWISS-Model-based homology modeling comparing SHP1-nSH2 interactions with pY+1 mutated PD-1 ITIM. Shown is a zoomed-in view of the hydrophobic pocket region with the pY+1 residue highlighted in a dashed oval. (**C**) Left: representative TIRF images of both PD-1 (GFP) and endogenous SHP1 (stained with anti-SHP1) in a supported lipid bilayer (SLB)-associated SHP2 KO Jurkat cell expressing indicated PD-1 variants. Right: dot plots summarizing anti-SHP1 fluorescence intensity (FI) normalized to GFP FI of 30-35 Jurkat cells under each condition recorded on the same day using the same microscope setting (see Materials and methods); Error bars: s.d. Scale bars: 5 µm. (**D**) Normalized anti-SHP1 FI for a subset of PD-1 variants shown in (C) plotted against the molecular volume of amino acids (AA) at the pY+1 position. (**E**) Relative IL-2 secretion by SHP2 KO Jurkat cells expressing indicated PD-1 variants. For each cell line, the relative IL-2 level was determined by normalizing the IL-2 level without pembrolizumab to that with 40 µg/ml (267 nM) pembrolizumab, which blocks PD-1 signaling. Error bars are s.d. from three independent coculture assays using three set of independently transduced cell lines, with each coculture assay run in technical duplicates. *p<0.05; **p<0.01; ***p<0.001; ns, not significant; Student's t-test.

The online version of this article includes the following source data and figure supplement(s) for figure 9:

**Source data 1.** Raw data for quantification graphs in *Figure 9C and E*.

**Figure supplement 1.** Flow cytometry histograms.

whether the induced SHP1 recruitment enhances PD-1 inhibitory function, we compared the pembrolizumab effect on IL-2 release from SHP2 KO Jurkat cells expressing similar levels of PD-1[WT], PD-1[G224A], PD-1[G224V], PD-1[G224L], PD-1[G224I], PD-1[G224F], or PD-1[G224W] (*Figure 9—figure supplement 1B*). Upon stimulation of these cells with Raji (PD-L1) cells with or without pembrolizumab, IL-2 ELISA showed that replacing glycine at pY+1 position with alanine, valine, leucine, or isoleucine, but not phenylalanine or tryptophan, significantly enhanced the PD-1-mediated inhibition of IL-2 secretion (*Figure 9E*). Taken together, these results revealed that a medium-sized nonpolar residue at pY+1 position of the ITIM is optimal for SHP1 recruitment.

## Discussion

SHP1 and SHP2 are key regulators of cell survival, proliferation, differentiation, and migration (*Gascoigne et al., 2015*; *Ke et al., 2007*; *Kuo et al., 2010*; *Lorenz, 2009*; *Paling and Welham, 2002*). Coexpressed in hematopoietic cells, they operate as central effectors for inhibitory immunoreceptors that contain ITIM and ITSM. Dissecting the precise mechanism by which these receptors discriminate between SHP1 and SHP2 is required to predict and understand their 'checkpoint' functions. In the present work, we combined biophysical, biochemical, and cellular imaging approaches to investigate the specificity dichotomy of PD-1 and BTLA. Our data have revealed distinct properties between the SH2 domains of SHP1 and SHP2, and between the ITIMs of these two checkpoint receptors. We also report that the differential SHP1-binding activities of PD-1 and BTLA can be largely attributed to a single (pY+1) residue of their ITIMs: the polarity and size of this residue gate SHP1 recruitment in T cells.

In human genome, at least 32 receptors contain two or more ITIMs or ITSMs in tandem. Conceivably, the tandem pY motifs favor their bivalent interactions with tSH2 of SHP1 and SHP2. Mathematical modeling predicts that bivalent binding, even contributed by two weak bonds, can produce stable protein complexes due to an avidity effect and a reduction of off-rate of protein complexes (*Diestler and Knapp, 2008*). In support of this notion, an earlier study showed that mutation of either ITIM or ITSM of BTLA abolishes its association with both SHP1 and SHP2 (*Watanabe et al., 2003*), a result that we have confirmed in the present study (*Figure 5B*). The bivalent binding mode appears to be less strict in the case of PD-1 since its ITIM mutant retained the ability to co-IP SHP2 (*Chemnitz et al., 2004*; *Okazaki et al., 2001*; *Patsoukis et al., 2020*; *Peled et al., 2018*; *Yokosuka et al., 2012*). A recent study suggested that SHP2 may crosslink two PD-1 molecules at the ITSM (*Patsoukis et al., 2020*), whereas other studies indicate that PD-1-ITIM contributes to SHP2 recruitment and activation (*Marasco et al., 2020*; *Peled et al., 2018*; *Yokosuka et al., 2012*). In our hands, ITIM mutant recruited significantly less SHP2 than PD-1[WT] in co-IP, microcluster enrichment, and single-molecule assays, supporting the involvement of ITIM, and the 1:1 bivalent mode as the dominant mode of PD-1:SHP2 interaction (*Figure 2*, *Figure 3—figure supplement 1*). In both BTLA and PD-1, the ITIM

tyrosine and ITSM tyrosine are separated by 25 residues (*Riley, 2009*), and similar spacing are found in numerous ITIM/ITSM-containing receptors (*Supplementary file 4*). This conserved distance may allow for simultaneous engagement of the ITIM and ITSM by the tSH2 of SHP1/SHP2 in the 1:1 bivalent mode. The bivalent binding likely occurs in a sequential fashion, in which the higher-affinity intermolecular contact precedes and converts the low-affinity binding to a pseudo-intramolecular event, as supported by mathematical modeling (*Zhou and Gilson, 2009*).

Our SPR data revealed certain promiscuities in the binding of their SH2 to ITIM and ITSM (*Figures 3 and 6*). Theoretically, SHP1 and SHP2 may bind to PD-1 and BTLA in either a parallel or an antiparallel orientation. However, free energy calculations suggest parallel mode as the most stable form for both PD-1 and BTLA (*Supplementary file 2*), consistent with a recent study on PD-1:SHP2 interaction (*Marasco et al., 2020*) and an earlier study on SHP2 interaction with PECAM1 (*Jackson et al., 1997*). To our knowledge, all ITIM/ITSM receptors with the exception of SLAMF5 in human genome contain an ITIM N-terminal to ITSM (*Supplementary file 4*). This spatial arrangement might allow these receptors to bind SHP1 and SHP2 in the parallel mode. The physiological significance of parallel binding is unclear, but it might increase the molecular reach of the PTPase domain (*Clemens et al., 2021*; *Zhang et al., 2019*), or avoid potential steric clash with the plasma membrane.

Extensive genetic and biochemical evidence shows that SHP1 and SHP2 differ in their physiological functions. SHP2 reportedly acts as both a positive and negative regulator, whereas SHP1 is primarily known as a negative regulator in cell signaling (*Lorenz, 2009*; *Poole and Jones, 2005*). The biochemical basis of their functional divergence is unknown. The most striking difference between SHP1 and SHP2, based on the current study, are their cSH2 domains: while SHP2-cSH2 binds to PD-1-ITSM with a high affinity, SHP1-cSH2 exhibits weak binding to PD-1-ITSM (*Figure 3A and B*). This distinction accounts for the undetectable PD-1:SHP1 association in T cells (*Xu et al., 2020*; *Yokosuka et al., 2012*), supported by our domain-swapping experiments (*Figure 4B and C*). Thus, PD-1 prefers SHP2 over SHP1 primarily based upon the stability of ITSM:cSH2 interaction. We speculate, in a more general sense, that the differing cSH2 domains of SHP1 and SHP2 might enable their recruitment to distinct signalosomes, leading to distinct functional outcomes. Indeed, SHP1-cSH2 and SHP2-cSH2 exhibit 51.9% homology in amino acid identities, and based on a recent NMR structure (*Marasco et al., 2020*), several residues implicated in PD-1-ITSM:SHP2-cSH2 interactions are altered in SHP1-cSH2 (*Figure 4A*).

Given the very weak affinity of the SHP1-cSH2 domain with the ITSM of both PD-1 and BTLA, we propose that receptors that stably recruit SHP1 must contain a strong docking site for SHP1-nSH2, such as the BTLA-ITIM (*Figure 6A and B*). The PD-1-ITIM, however, might be too weak to support stable, bivalent PD-1:SHP1 binding. Moreover, because SHP1/SHP2 undergo autoinhibition due to intramolecular (cis) interactions between the nSH2 and the catalytic domain (*Pádua et al., 2018*; *Pei et al., 1994*; *Yang et al., 2003*), a stronger ITIM might also allow the receptor to more efficiently release the autoinhibition of SHP1/SHP2 in trans. This may also help explain the inability of SHP1 to compensate for PD-1 function in SHP2-deficient T cells (*Figure 1C*; *Xu et al., 2020*), even though in a subset of Jurkat cells, we were able to detect weak SHP1 enrichment to PD-1 microclusters.

Biochemical research on PD-1 has led to a consensus that ITSM is its primary docking site for SHP2 (*Chemnitz et al., 2004*; *Okazaki et al., 2001*; *Patsoukis et al., 2020*; *Peled et al., 2018*; *Yokosuka et al., 2012*). Along this line, the current study shows that PD-1 uses ITSM to prefer SHP2 over SHP1 (*Figure 3A and B*). Thus, ITSM is the 'dominant hand' of PD-1. In contrast, the 'dominant hand' of BTLA appears to be its ITIM. Our data show that ITIM of BTLA serves as the primary docking site for SHP1 and allows BTLA to discriminate between SHP1 and SHP2 (*Figure 6A and B*).

SH2 binding is contributed by both pY and its flanking residues (*Kuriyan and Cowburn, 1997*; *Pawson, 1995*). Alanine is conserved at pY+1 position in the ITIMs of numerous immunoreceptors (*Figure 8—figure supplement 1B*), suggesting its critical role in the physiological functions of these receptors. In this sense, human PD-1 is an interesting exception: its ITIM appears to be the only ITIM that has a glycine at pY+1 position (*Figure 8—figure supplement 1B*). Our study demonstrates that the pY+1 glycine inhibits SHP1 recruitment, and replacement of glycine to an alanine, valine, leucine, or isoleucine was sufficient to induce PD-1:SHP1 association in T cells, whereas a polar, charged, or bulky residue was significantly less efficient to promote SHP1 binding (*Figures 8 and 9*). These results suggest that a medium-sized nonpolar residue at pY+1 position is a defining feature of a SHP1-docking site, specifically its nSH2 domain. Consistent with this model, PECAM1, another receptor

reported to strongly prefer SHP2 over SHP1 (*Jackson et al., 1997*; *Sagawa et al., 1997*), has a polar residue threonine at pY+1 of its ITIM.

How could a medium-sized hydrophobic residue at pY+1 contribute to SH2 binding? Our homology structural modeling reveals a hydrophobic pocket within SHP1-nSH2 that best coordinates a medium-sized nonpolar side chain from the pY+1 position of ITIM (*Figure 9A and B*). We speculate that similar hydrophobic contacts likely stabilize BTLA-ITIM:SHP1-nSH2 interactions. A glycine at pY+1, as occurs in human PD-1-ITIM, likely impairs such hydrophobic interactions due to its small size or its structural flexibility, whereas a bulky residue (phenylalanine or tryptophan) would inhibit the interaction due to a steric effect. Further structural studies will be needed to definitively test these notions.

# Materials and methods

## Key resources table

| Reagent type (species) or resource | Designation | Source or reference | Identifiers | Additional information |
|---|---|---|---|---|
| Antibody | Biotin anti-human CD3 $\varepsilon$ (OKT3, mouse monoclonal) | BioLegend | Cat#: 317320; RRID:AB_10916519 | TIRF (1:100) |
| Antibody | Anti-phosphotyrosine (pY20, mouse monoclonal) | Sigma-Aldrich | Cat#: P4110-1MG; RRID:AB_477342 | WB (1:500) |
| Antibody | Anti-GFP (rabbit polyclonal) | Invitrogen | Cat#: A-6455; RRID:AB_221570 | WB (1:1000) |
| Antibody | Anti-SHP1 (rabbit polyclonal) | Proteintech | Cat#: 24546-1-AP; RRID:AB_2879600 | WB (1:1000) |
| Antibody | Anti-SHP1 (rabbit monoclonal) | Life Technologies | Cat#: 3H20L13; RRID:AB_2809241 | IF (1:100) |
| Antibody | Anti-SHP2 (mouse monoclonal) | Santa Cruz Biotechnology | Cat#: sc-7384; RRID:AB_628252 | WB (1:200) |
| Antibody | Anti-SHP2 (mouse monoclonal) | BD Biosciences | Cat#: 610622; RRID:AB_397954 | IF (1:200) |
| Antibody | Anti-GAPDH (rabbit polyclonal) | Proteintech | Cat#: 10494-1-AP; RRID:AB_2263076 | WB (1:1000) |
| Antibody | F(ab')2-goat anti-rabbit IgG (H + L) secondary antibody, Alexa Fluor 568 | Invitrogen | Cat#: A21069; RRID:AB_2535730 | IF (1:1000) |
| Antibody | PE anti-human BTLA (MIH26) | BioLegend | Cat#: 344505; RRID:AB_2043945 | Flow (1:100) |
| Antibody | PE anti-human PD-1 (MIH4) | Thermo Fisher | Cat#: 12-9969-42; RRID:AB_10736473 | Flow (1:100) |
| Antibody | Pacific Blue anti-human PD-1 (EH12.2H7) | BioLegend | Cat#: 329916; RRID:AB_2283437 | Flow (1:100) |
| Antibody | Pembrolizumab (anti-human PD-1, IgG4) | Selleck Chemicals | Cat#: A2005 | |
| Commercial assay or kit | Human IL-2 ELISA kit | Thermo Fisher | Cat#: 88702577; RRID:AB_2574952 | |
| Recombinant protein | Human PD-L1-His | Sino Biological | Cat#: 10084-H08H | |
| Recombinant protein | Human ICAM-1-His | Sino Biological | Cat#: 10346-H08H | |
| Recombinant protein | Streptavidin | Invitrogen | Cat#: S888 | |
| Recombinant protein | SEE super antigen | Toxin Technologies | Cat#: ET404 | |
| Chemical compound, drug | Paraformaldehyde | Fisher Scientific | Cat#: 50980494 | |
| Chemical compound, drug | 100× penicillin-streptomycin | GE Healthcare | Cat#: SV30010 | |
| Chemical compound, drug | Polyethylenimine (PEI) | Fisher Scientific | Cat#: NC1014320 | |
| Chemical compound, drug | 1,2-Dioleyl-sn-glycero-3-phosphocholine (POPC) | Avanti Polar Lipids | Cat#: 850457C | |

*Continued on next page*

*Continued*

| Reagent type (species) or resource | Designation | Source or reference | Identifiers | Additional information |
|---|---|---|---|---|
| Chemical compound, drug | 1,2-Dioleoyl-sn-glycero-3-[(N-(5-amino-1-carboxypentyl) iminodiacetic acid) succinyl] nickel salt (DGS-NTA-Ni) | Avanti Polar Lipids | Cat#: 790404C | |
| Chemical compound, drug | 1,2-Dipalmitoyl-sn-glycero-3-phosphoethanolamine-N-(biotinyl) (sodium salt) | Avanti Polar Lipids | Cat#: 870285P | |
| Chemical compound, drug | 1,2-Dioleoyl-sn-glycero-3-phosphoethanolamine-N-[methoxy(polyethylene glycol)–5000] ammonium salt (PEG 5000-PE) | Avanti Polar Lipids | Cat#: 880230C | |
| Chemical compound, drug | Imidazole | Sigma-Aldrich | Cat#: I202 | |
| Chemical compound, drug | TCEP-HCl | Gold Biotechnology | Cat#: TCEP10 | |
| Chemical compound, drug | Alexa Fluor 647 NHS ester | Thermo Fisher | Cat#: A37573 | |
| Chemical compound, drug | SNAP ligand-JF549 | Janelia Research Campus (HHMI) | PMID:28924668 | |
| Chemical compound, drug | SNAP ligand-JF646 | Janelia Research Campus (HHMI) | PMID:28924668 | |
| Chemical compound, drug | Hellmanex III | Sigma | Cat#: Z805939-1EA | |
| Chemical compound, drug | Biotin | Sigma-Aldrich | Cat#: B4501 | |
| Chemical compound, drug | ATP | Gold Biotech | Cat#: A-081-100 | |
| Chemical compound, drug | Ni-NTA resin | Thermo Fisher | Cat#: 88223 | |
| Chemical compound, drug | GFP-Trap | Chromotek | Cat#: gta-20 | |
| Chemical compound, drug | Glutathione agarose resin | Gold Biotechnology | Cat#: G-250-50 | |
| Chemical compound, drug | Zeba Spin Desalting Columns | Thermo Fisher | Cat#: 89890 | |
| Chemical compound, drug | Gel Filtration Standard | Bio-Rad | Cat#: 1511901 | |
| Chemical compound, drug | DMEM High Glu w/Gln w/o Pyr | Thermo Fisher | Cat#: MT10017CV | |
| Chemical compound, drug | RPMI 1640, w/Gln and 25 mM HEPES | Corning | Cat#: MT 10-041CM | |
| Chemical compound, drug | Fetal bovine serum, heat-inactivated | Omega Scientific | Cat#: FB-02 | |
| Other | Glass-bottomed 96-well plate | DOT Scientific Inc | Cat#: MGB096-1-2-LG-L | |
| Other | Ni sensor chip | Nicoya | Cat#: SEN-AU-100-10-NTA | |
| Cell line (*Homo sapiens*) | Jurkat E6.1 | Provided by Dr. Arthur Weiss (University of California San Francisco) | RRID:CVCL_0065 | |
| Cell line (*H. sapiens*) | Raji | Provided by Dr. Ronald Vale (University of California San Francisco) | RRID:CVCL_0511 | |
| Cell line (*H. sapiens*) | HEK-293T | Provided by Dr. Ronald Vale (University of California San Francisco) | RRID:CVCL_0063 | |

*Continued*

| Reagent type (species) or resource | Designation | Source or reference | Identifiers | Additional information |
|---|---|---|---|---|
| Cell line (*H. sapiens*) | SHP1 KO Jurkat | *Xu et al., 2020* | PMID:32437509 | |
| Cell line (*H. sapiens*) | SHP2 KO Jurkat | *Xu et al., 2020* | PMID:32437509 | |
| Cell line (*H. sapiens*) | SHP1/2 DKO Jurkat | *Xu et al., 2020* | PMID:32437509 | |
| Cell line (*H. sapiens*) | Raji (PD-L1-mCherry) | *Xu et al., 2020* | PMID:32437509 | |
| Cell line (*H. sapiens*) | Raji (HVEM-mRuby2) | *Xu et al., 2020* | PMID:32437509 | |
| Cell line (*H. sapiens*) | SHP2 KO Raji (PD-L1-mCherry) | *Xu et al., 2020* | PMID:32437509 | |
| Cell line (*H. sapiens*) | SHP2 KO Raji (HVEM-mRuby2) | *Xu et al., 2020* | PMID:32437509 | |
| Software, algorithm | FlowJo V10 | TreeStar | RRID:SCR_008520 | Flow data processing and analysis |
| Software, algorithm | GraphPad Prism v8 | GraphPad | RRID:SCR_002798 | Graphs and statistical analysis |
| Software, algorithm | ImageJ | NIH | RRID:SCR_003070 | Image acquisition, processing, and analysis |
| Software, algorithm | Fiji | MPI-CBG | PMID:22743772; RRID:SCR_002285 | Image processing and analysis |
| Software, algorithm | Micro-Manager | GitHub | PMID:25606571; RRID:SCR_000415 | |
| Software, algorithm | ThunderStorm | GitHub | PMID:24771516; RRID:SCR_016897 | Single-molecular images analysis |
| Software, algorithm | OpenSPR | Nicoya | N/A | SPR data acquisition |
| Software, algorithm | TraceDrawer | Ridgeview Instruments | N/A | SPR data analysis |
| Software, algorithm | PyMOL | Schrödinger, Inc | RRID:SCR_000305 | Structural modeling, simulation |

## Cell lines and cultures

Jurkat E6.1 cells were provided by Dr. Arthur Weiss (University of California San Francisco). HEK293T cells and Raji cells were provided by Dr. Ronald Vale (University of California San Francisco). SHP1 KO Jurkat, SHP2 KO Jurkat, SHP1/2 DKO Jurkat, Raji (PD-L1-mCherry), Raji (HVEM-mRuby2), SHP2 KO Raji (PD-L1-mCherry), and SHP2 KO Raji (HVEM-mRuby2) cells were generated in our previous study (*Xu et al., 2020*). Each gene of interest was introduced into Jurkat cells via lentiviral transduction, as described previously (*Xu et al., 2020*). Briefly, each cDNA was cloned into a pHR vector backbone, and co-transfected with pMD2.G and psPAX2 packaging plasmids into HEK293T cells using polyethylen-imine (PEI, Fisher Scientific, #NC1014320). Virus-containing supernatants were harvested at 60–72 hr post-transfection. WT Jurkat cells, SHP1 KO Jurkat cells, SHP2 KO Jurkat cells, or SHP1/2 DKO Jurkat cells were resuspended with the desired virus supernatant, centrifuged at 35°C, 1000× g for 30 min, and incubated overnight at 37°C 5% $CO_2$ before replacing the virus supernatant with complete RPMI-1640 medium. HEK293T cells were maintained in DMEM medium (Genesee Scientific, #25-501) supplemented with 10% fetal bovine serum (Omega Scientific, #FB-02) and 1% 100× penicillin-streptomycin (GE Healthcare, #SV30010) at 37°C/5% $CO_2$. Jurkat and Raji cells, authenticated by ATCC using short tandem repeats (STR) profiling, were maintained in RPMI-1640 medium (Corning, #10-041CM) supplemented with 10% fetal bovine serum, 100 U/ml of penicillin, and 100 µg/ml of streptomycin at 37°C/5% $CO_2$. The lack of mycoplasma in the cell lines was confirmed using PCR Mycoplasma Detection Kit (Applied Biological Materials Inc, G238). Cells used in the present study were used within 10 passages from thawing.

## Recombinant proteins

For SPR assays in *Figures 3A and 6A*, human BTLA$^{ICD}$ (aa 190–289) and PD-1$^{ICD}$ (aa 194–288) tyrosine mutants (BTLA$^{FFYF}$, BTLA$^{FFFY}$, PD-1$^{FY}$, PD-1$^{YF}$) were expressed with an N-terminal His$_{10}$ tag in *Escherichia coli* using the pET28A vector and purified using Ni-NTA agarose (Thermo Fisher, #88223) as described (*Xu et al., 2020*). His$_{10}$ tagged human PTK Fyn was expressed in the Bac-to-Bac baculovirus system

and purified using Ni-NTA agarose. SNAP-SHP1-nSH2, SNAP-SHP1-nSH2, SNAP-SHP2-nSH2, and SNAP-SHP2-cSH2 were expressed with an N-terminal GST tag followed by a PreScission recognition sequence (LEVLFQGP), in *E. coli* via the pGEX6p-2 vector. For the single-molecule imaging assay in *Figure 3—figure supplement 1*, all proteins were expressed with an N-terminal GST tag followed by a PreScission recognition sequence (LEVLFQGP), in *E. coli* via the pGEX6p-2 vector. These included SNAP-SHP1-tSH2, SNAP-SHP2-tSH2, as well as the ICD of PD-1 WT or its tyrosine mutants fused with an N terminal Avi-tag (GLNDIFEAQKIEWHE) and a C-terminal SNAPf tag (Avi-PD-1$^{YY}$-SNAPf, Avi-PD-1$^{FY}$-SNAPf, Avi-PD-1$^{YF}$-SNAPf). All GST fusion proteins were purified using Glutathione Agarose 4B (Gold Biotechnology, #G-250-50), and eluted with HEPES buffered saline (HBS, 50 mM HEPES, 150 mM NaCl, 0.5 mM TCEP [Gold Biotechnology, #TCEP10], pH 7.5) containing 20 units/ml 3C protease to remove the GST tag. After elution, SNAP-SHP1-tSH2 and SNAP-SHP2-tSH2 were further labeled with SNAP ligand-JF646 (Janelia Research Campus [HHMI], #Janelia 2014-013) at 4°C overnight. Avi-PD-1$^{YY}$-SNAPf, Avi-PD-1$^{FY}$-SNAPf, and Avi-PD-1$^{YF}$-SNAPf were further labeled with SNAP ligand-JF549 (Janelia Research Campus [HHMI], #Janelia 2014-013) and biotin in the presence of 1 mM biotin (Sigma-Aldrich, #B4501), 1 μM BirA, and 10 mM ATP (Gold Biotech, #A-081-100) at 4°C overnight. All affinity-purified proteins were subjected to gel filtration chromatography using HBS containing 10% glycerol and 1 mM TCEP. The monomer fractions were pooled, snap frozen, and stored at –80°C in small aliquots. Gel filtration standards (Bio-Rad, #1511901) were run to confirm the sizes of eluted proteins.

## Phosphorylation of recombinant PD-1 and BTLA ICD proteins

Purified His$_{10}$-PD-1$^{ICD}$, His$_{10}$-BTLA$^{ICD}$, and Avi-PD-1$^{ICD}$-SNAPf proteins were incubated with 50 nM purified Fyn, 2 mM ATP, and 10 mM Na$_3$VO$_4$ at room temperature (RT) for 6 hr to achieve full phosphorylation, as indicated by the complete shift of electrophoretic mobility on sodium dodecyl sulfate polyacrylamide gel electrophoresis (SDS-PAGE). Monomeric form of these pre-phosphorylated proteins was purified using gel filtration chromatography in HBS containing 10% glycerol and 1 mM TCEP, and stored in –80°C in aliquots.

## SHP2 antibody labeling

For fluorescent staining of SHP2 in T cells, anti-SHP2 was fluorescently labeled using Alexa Fluor 647 NHS ester (Thermo Fisher, #A37573), and unreacted chemicals were removed using Zeba Spin Desalting Columns (Thermo Fisher, #89890) following the manufacturer's instructions.

## Jurkat-Raji coculture assay for immunoprecipitation and IL-2 secretion

For *Figures 1, 2A, 4C, 5B, 7B, 8B and E*, *Figure 5—figure supplement 1B*, *Figure 7—figure supplement 2B*, Jurkat cells were starved in serum-free RPMI medium at 37°C for 3 hr prior to coculture. Raji cells were preincubated with 30 ng/ml SEE (Toxin Technologies, #ET404) in RPMI medium for 30 min at 37°C. In order to avoid SHP2 competition from Raji cells, SHP2 KO Raji cells were used in *Figures 4C, 7B, 8B and E*, *Figure 7—figure supplement 2B*. Afterward, 4 million SEE-loaded Raji cells and 4 million Jurkat cells were precooled on ice and mixed in a 96-well plate. After centrifugation at 300× g for 1 min at 4°C to initiate Raji-Jurkat contact, cells were immediately transferred to a 37°C water bath. At 5 min or the time points indicated in the figures, Raji-Jurkat conjugates were lysed with HBS containing 5% glycerol, detergent (1% NP-40), protease inhibitor (1 mM PMSF), and phosphatase inhibitors (10 mM Na$_3$VO$_4$ and 10 mM NaF). GFP-tagged PD-1 variants or BTLA variants were IP'ed from the lysate using GFP-Trap (Chromotek, #gta-20). Equal fractions of the IP samples were subjected to SDS-PAGE and blotted with indicated antibodies. For IL-2 assays shown in *Figures 1C and 9E*, Raji B cells were preincubated with 30 ng/ml SEE in RPMI medium for 30 min at 37°C. Jurkat cells were preincubated with indicated concentrations of pembrolizumab or with PBS at RT for 30 min. Afterward, 1 × 10$^5$ SEE-pulsed Raji B cells and 2 × 10$^5$ pembrolizumab/PBS-conditioned Jurkat T cells were mixed in a 96-well U-bottom plate in triplicate wells, followed by centrifugation at 300× g for 1 min to initiate cell-cell contact. Cultures were then incubated in a 37°C/5% CO$_2$ incubator, 6 hr later, supernatants were harvested and IL-2 levels were measured by an ELISA kit (Thermo Fisher, #88702577). For each replicate of each cell line, the measured IL-2 levels were normalized to the condition with the highest IL-2 level and shown as relative IL-2 levels.

## Flow cytometry

Flow cytometry was conducted in an LSRFortessa cell analyzer (BD Biosciences). Indicated Jurkat cells were washed with PBS and analyzed after staining with PE anti-human BTLA (MIH26), PE anti-human PD-1 (MIH4), or Pacific Blue anti-human PD-1 (EH12.2H7). For *Figure 4—figure supplement 1*, mCherry levels were measured using a FACSAria cell sorter (BD Biosciences) due to a lack of 561 nm laser in the LSRFortessa cell analyzer. Data were analyzed using FlowJo (FlowJo, LLC).

## SLB preparation and functionalization

SLBs were formed on glass-bottomed 96-well plate (DOT Scientific Inc, #MGB096-1-2-LG-L). Briefly, the plate was cleaned with 2.5% Hellmanex (Sigma-Aldrich, #Z805939-1EA), etched with 5 N NaOH, and used for SLB formation as previously described (*Ahrends et al., 2017*). Briefly, small unilamellar vesicles (SUVs) derived from dried lipid film containing 95.5% POPC (Avanti Polar Lipids, #850457C), 2% biotin-DPPE (Avanti Polar Lipids, #870285P), 2% DGS-NTA-Ni (Avanti Polar Lipids, #790404C), and 0.1% PEG 5000-PE (Avanti Polar Lipids, #880230C) were added onto freshly treated plates to form SLBs. The SLBs were rinsed with wash buffer (1× PBS containing 0.1% BSA) and mixed with 1 μg/ml streptavidin, 0.1 nM His-tagged human PD-L1 ECD, and 3 nM His-tagged human ICAM-1 ECD at 37°C for 1 hr. Afterward, the SLBs were rinsed with wash buffer and further incubated with 5 μg/ml biotin anti-human-CD3 $\varepsilon$ at 37°C for 30 min, followed by three rinses with wash buffer and three rinses with imaging buffer (20 mM HEPES pH 7.5, 137 mM NaCl, 5 mM KCl, 1 mM $CaCl_2$, 2 mM $MgCl_2$, 0.7 mM $Na_2HPO_4$, 6 mM D-glucose).

## Cell-SLB image acquisition and analysis

Jurkat cells were resuspended in imaging buffer and overlaid onto freshly formed PD-L1/ICAM/Okt3-functionalized SLBs. After 5 min incubation at 37°C, SLB-bound cells were overlaid with 2% paraformaldehyde (PFA, Fisher Scientific, #50980494), and incubated at RT for 15 min for fixation. SLB-associated, PFA-treated cells were washed with blocking buffer (1× PBS containing 3% BSA), and permeabilized with 1× PBS containing 3% BSA and 0.1% Saponin at RT for 30 min. To observe mCherry-tagged SHP1$^{WT}$ or domain swapping mutants (*Figure 4B*), cells were directly imaged at both GFP (488 nm) and mCherry (561 nm) channels. To observe endogenous SHP2 (*Figure 2B*), the permeabilized cells were stained with Alexa-Fluor-647-labeled anti-SHP2 at 4°C for 16 hr, followed by fixation with 4% PFA. To observe endogenous SHP1 (*Figures 7C, 8C, F and 9C*), the permeabilized cells were stained with anti-SHP1 at 4°C for 16 hr, followed by fixation with 4% PFA, and further staining with Alexa-Fluor-568-labeled anti-rabbit IgG at RT for 1 hr and another treatment with 4% PFA. The fluorescent cell images were acquired on a Nikon Eclipse Ti TIRF microscope equipped with a 100× Apo TIRF 1.49 NA objective lens, controlled by the Micro-Manager software (*Edelstein et al., 2014*). Fiji (*Schindelin et al., 2012*) was used to quantify the degree of recruitment of SHP1 and SHP2 to PD-1 microclusters. Mask images identifying the area of PD-1 microclusters were generated by applying the 'subtract background' command to PD-1 (mGFP) images using the default setting. The fluorescent signals of anti-SHP1, anti-SHP2, and PD-1 (GFP) in the masked overlaid images were measured and used to calculate the anti-SHP1 FI/GFP FI ratio and SHP2/PD-1 ratio for each cell.

## SPR assay

For *Figures 3A and 6A*, direct interaction between individual SH2 (nSH2 or cSH2) of SHP1 or SHP2 and BTLA ICD, PD-1 ICD tyrosine mutants (BTLA$^{FFYF}$, BTLA$^{FFFY}$, PD-1$^{FY}$, PD-1$^{YF}$) was monitored by OpenSPR (Nicoya) equipped with Ni sensor chip (Nicoya, #SEN-AU-100-10-NTA). Pre-phosphorylated His$_{10}$-tagged PD-1 ICD or BTLA ICD was immobilized onto a Ni sensor chip to achieve approximately 1500 RU by following the Ni sensor wizard in OpenSPR software. SH2 of interest was diluted in running buffer (20 mM HEPES, 150 mM NaCl, 5 mM imidazole [Sigma-Aldrich, #I202], 0.05% Tween-20, 10% glycerol, pH 7.5), and injected. The association and dissociation phases of SH2 were monitored at a flow rate of 20 μl/min. The Ni sensor chip was regenerated with 50 mM NaOH before injecting the next SH2. Sensorgrams were analyzed using the 'Evaluate EC50' method in TraceDrawer software (Ridgeview Instruments).

## Single-molecule imaging assay

For *Figure 3—figure supplement 1*, surface-passivated coverslips and slide glasses used in single-molecule imaging assays were prepared and assembled as previously described (*Chandradoss et al., 2014*). Surface-passivated glass chambers were incubated with blocking buffer (50 mM HEPES, 150 mM NaCl, 0.1% BSA) at RT for 5 min and with 0.5 µM streptavidin in blocking buffer at RT for 5 min, followed by two washes with blocking buffer. The glass chambers were further incubated with 10 pM pre-phosphorylated, biotinylated, JF549-labeled PD-1$^{YY}$, PD-1$^{FY}$, or PD-1$^{YF}$, and incubated at RT for 5 min. The unbound proteins were removed with blocking buffer and JF646-labeled SH2 proteins were injected to the chambers. The fluorescent images were acquired at 20 Hz on a Nikon Eclipse Ti TIRF microscope equipped with a 100× Apo TIRP 1.49 NA objective lens, controlled by the Micro-Manager software (*Edelstein et al., 2014*).

## Fluorescent intensity distribution analysis and photobleaching analysis

For *Figure 3—figure supplement 1C and D*, JF646-labeled PD-1 attached on coverslips were illuminated with the 50 mW 488 nm laser for photobleaching observation. The initial positions of fluorescent spots for JF646-labeled PD-1 were determined by the 'ThunderSTORM' plugin (*Ovesný et al., 2014*) in Fiji. The fluorescent intensities at each determined position in initial images were measured and plotted into histogram fit with Gaussian distributions, and those of whole image stacks were measured for fluorescent trajectories.

## Fluorescent trajectory analysis and colocalization analysis

For *Figure 3—figure supplement 1D and F*, the positions of fluorescent spots for JF549-labeled PD-1 were determined by the 'ThunderSTORM' plugin in Fiji. Likewise, the fluorescent movies for JF646-labeled tSH2 proteins were Z-projected to 'maximum intensity' and analyzed by the 'ThunderSTORM' plugin to determine the positions where tSH2 proteins bound. The positions detected in both JF549 and JF646 channels were determined as spots representing PD-1 molecules that recruited tSH2 protein at any given time during the image acquisition. The fluorescent intensities of JF646 within those areas were measured to fluorescent trajectories of tSH2 proteins. The bound and unbound states of tSH2 proteins in the fluorescent trajectories were estimated with Hidden Markov Model in a custom written Python script (https://github.com/HuiLabUCSD/Xu-and-Masubuchi-et-al-eLife-2021; *Masubuchi, 2021*; copy archived at swh:1:rev:48368d5a6f69d8e68ffcb1a3fd67a7d50219015f). The colocalization rates of PD-1 and tSH2 protein were calculated by dividing the total number of PD-1 spots by the number of PD-1 spots that recruited SH2 proteins within 0.5 s of image acquisitions.

## Structural modeling and simulation

For *Figure 9A and B*, structural modeling of SHP1-nSH2 interaction with WT or mutant PD-1-ITIM was generated by homology modeling using the SWISS-Model (*Waterhouse et al., 2018*) based on the reported complex structure of SHP2-nSH2 and PD-1-ITIM (PDB: 6ROY) (*Marasco et al., 2020*). *Figures 4A, 9A and B* were prepared using PyMOL (http://www.pymol.org/).

## Quantification and statistical analysis

Data were shown as mean ± s.d., and the number of replicates is indicated in figure legends. Curve fitting and normalization were performed in GraphPad Prism 8 (GraphPad). Statistical significance was evaluated by either Student's t-test or two-way ANOVA test (*$p<0.05$; **$p<0.01$; ***$p<0.001$; ****$p<0.0001$; ns, not significant). Data with $p<0.05$ are considered statistically significant.

# Acknowledgements

We thank J Wilhelm (UCSD) for sharing the TIRF microscope; P Dennett and J Zhang for critically reading the manuscript. TM is supported by the Human Frontier Science Program postdoctoral fellowship. This work was supported by R37 CA239072 from the National Institute of Health, a Searle Scholar Award from the Kinship Foundation, and a Pew Biomedical Scholar Award from the Pew Charitable Trusts to EH.

## Additional information

### Funding

| Funder | Grant reference number | Author |
|---|---|---|
| National Cancer Institute | R37 CA239072-03 | Enfu Hui |
| Pew Charitable Trusts | | Enfu Hui |
| Searle Scholars Program | | Enfu Hui |

The funders had no role in study design, data collection and interpretation, or the decision to submit the work for publication.

### Author contributions

Xiaozheng Xu, designed the project; generated plasmids, proteins and cell lines; conducted the co-IP experiments and IL-2 assays; initiated the collaboration with Qixu Cai on structural modeling and simulations; drafted, revised and finalized the manuscript; Takeya Masubuchi, designed the project; generated plasmids, proteins and cell lines; conducted the cell-SLB assays, SPR experiments, and single molecule imaging; sequence queries of ITIM/ITSM containing receptors, wrote the customized code for single molecule imaging analyses; drafted, revised and finalized the manuscript; Qixu Cai, conducted the structural modeling and simulations; Yunlong Zhao, generated plasmids, proteins and cell lines; Enfu Hui, conceived and supervised the project; drafted, revised and finalized the manuscript; obtained the funding

### Author ORCIDs

Xiaozheng Xu 
Takeya Masubuchi 
Qixu Cai 
Yunlong Zhao 
Enfu Hui 

### Decision letter and Author response

Decision letter https://doi.org/10.7554/eLife.74276.sa1
Author response https://doi.org/10.7554/eLife.74276.sa2

## Additional files

### Supplementary files

• Supplementary file 1. $K_d$ values of interactions between individual SH2 of SHP1/SHP2 and phosphorylated ITIM/ITSM of PD-1/BTLA; mean ± s.d. (n = 3).

• Supplementary file 2. Table summarizing ΔG of PD-1/BTLA:SHP1/SHP2-tSH2 interactions in a parallel or an anti-parallel mode.

• Supplementary file 3. Table summarizing ΔG of individual SH2:ITIM/ITSM interactions.

• Supplementary file 4. A list of immunoreceptors that contain both ITIM and ITSM.

• Transparent reporting form

### Data availability

All data generated or analysed during this study are included in the manuscript and supporting file.

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
