## [Editor Report]

This study elegantly addressed the SHP1/SHP2 preferences of ITIM/ITSM-containing inhibitory immunoreceptors PD-1 and BTLA, with solid evidence from cell-based, biochemical, biophysical, and domain-swapping assays. Importantly, it lays the foundation for further structural, physiological, and therapeutic studies.

---

## [Decision Letter]

[Editors' note: this paper was reviewed by Review Commons.]

---

## [Author Response]

Reviewer #1 (Evidence, reproducibility and clarity (Required)):This paper explores an interesting problem of SHP1/SHP2 preferences of inhibitory immunoreceptors. The author are quick to point out that many of their individual data points confirm published results at some level, but the power of the paper is in the parallel analysis of both PD1, which is strongly biased towards SHP2 and BTLA, which is biased towards SHP1. This gives them the opportunity to test the predictions of descriptive experiment by making simple mutated receptors with swapped ITIM or ITSM domains.The work is very well done and generally the authors are quite careful and precise about the language used to describe results, in general.The results are quite striking in that the find plenty of evidence for transient interaction of SHP1 with PD1 based on the biophysical measurements, but don't detect the interactions in pull down or in "in cell" microcluster recruitment experiments. In describing the pull-downs they discuss the issue of dissociation during washing potentially missing interactions that are taking place. I would prefer that the pull down is fine evidence for binding, but lack of pull down is not evidence for lack of binding. They should double check that this language is consistent. Also, unless something has changed in the microcluster binding experiments, this in situ recruitment of SHP2 to PD1 is only observed or a 2-3 minutes and then can't be detected, the situation for SHP2 becoming the same as it is for SHP1. If the kinetics are different in the cleaner systems that have now developed they should show this in a primary figure as this would be then different when what is reported previously.

We agree with the reviewer that pull down is evidence for binding. Indeed, in most, if not all of our assays, our results with pull down were consistent with those in the microcluster imaging. It does seem though the differences observed in the “in cell” microcluster recruitment experiments are less striking than in the pull down experiments. As suggested by the reviewer, we have checked through the manuscript and ensure the language is accurate and consistent.

In our recent study (PMID: 32437509), we conducted a side-by-side comparison of SHP2 and SHP1 recruitment kinetics to PD-1 in a similar system. Both microcluster imaging and co-IP assays showed that PD-1:SHP2 association lasted at least 10 minutes, whereas PD-1:SHP1 recruitment was nearly undetectable. The duration of PD-1:SHP2 association was in good agreement with Takashi Saito’s finding in CD4^+^ mouse primary T cells (PMID: 22641383). Regardless the somewhat different kinetics in different studies, SHP2 recruitment was transient, as pointed out by the reviewer. We believe that some other effectors contribute to PD-1 inhibitory signaling. In supportive of this notion, we recently found that PD-1 remains partially inhibitory in CD8^+^ T cells deficient in both SHP1 and SHP2 (PMID: 32437509).

The gap in this study is lack of any functional analysis. The Jurkat model could be quite useful as they have a relatively clean system for asking if the transient binding of SHP1 to PD1 has any functional impact, which they have not yet followed through on. Does PD-1 recruited SHP2 have any impact on function after the 5 minutes? Furthermore, the authors need to keep in mind that mice deficient in SHP2 respond to anti-PD1 checkpoint therapies (Rota, G., Niogret, C., Dang, A. T., Barros, C. R., Fonta, N. P., Alfei, F., Morgado, L., Zehn, D., Birchmeier, W., Vivier, E., and Guarda, G. (2018). Shp-2 Is Dispensable for Establishing T Cell Exhaustion and for PD-1 Signaling in vivo. Cell Rep, 23(1), 39-49. https://doi.org/10.1016/j.celrep.2018.03.026). This is an important issue to discuss in light the very interesting binding analysis the authors have performed. But I think the functional analysis can be part of a future paper.

We appreciate the reviewer’s comments and suggestions. To address the concern on the functional relevance, we first tested the magnitude of PD-1 mediated IL-2 inhibition in WT, SHP1 KO, SHP2 KO and SHP1/SHP2 double KO Jurkat cells using pembrolizumab, an FDAapproved PD-1 blockade antibody, to titrate PD-1 signaling. This new experiment revealed that deletion of SHP1 from Jurkat cells had little effect on PD-1 mediated suppression of IL-2. In contrast, deletion of SHP2 from Jurkat cells significantly decreased the PD-1 inhibitory effect. These results strongly suggest that the transient binding of SHP1 to PD1 does not contribute significantly to PD-1 signaling. We now show this new data as Figure 1C of the revised manuscript.

Relative IL-2 levels produced by PD-1+ WT, SHP1 KO, SHP2 KO or SHP1/2 DKO Jurkat cells stimulated with PD-L1+ Raji cells in the presence of increasing concentrations of pembrolizumab (0, 0.4, 1.3, 4.4, 13.3, 44, 133, or 267 nM). For each type of Jurkat cells, IL-2 data was normalized to that at the highest pembrolizumab concentration (267 nM, or 40 μg/ml). Error bars are s.d. from three independent experiments. ****P < 0.0001; ns, not significant; two-way ANOVA test.

Moreover, we conducted another functional assay to test whether SHP1 recruitment, induced by mutations at the PD-1-ITIM pY+1 residue, correlates with the abilities of PD-1 mutants to suppress IL-2 production. Through careful titration of lentivirus titer, we were able to establish seven Jurkat cell lines, each expressing similar levels of a PD-1 variant, with a different nonpolar residues at the pY+1 position of ITIM (see Figure S1H). After stimulating these cell lines side-by-side with PD-L1+ Raji B cells, we found a strong correlation between the degree of SHP1 recruitment and the degree of IL2 suppression, see Figure 9E of the revised manuscript. Therefore, our new functional data demonstrate that the pY+1 residue of ITIM gates SHP1 recruitment, but also influence PD-1 inhibitory function.

We also agree with this reviewer that our manuscript would benefit from discussion of functional analysis. To this end, we have rewritten both the introduction and Discussion section extensively to discuss more on the physiological and functional relevance.

Speaking of the SHP1/SHP2 recruitment kinetics, as the reviewer alluded to, we did observe SHP2 dissociation from PD-1 after 10 minutes, as seen by Saito and colleagues (PMID: 22641383), as well as our recent study (PMID: 32437509). On the other hand, there is also clear evidence that SHP2 is required for long term inhibitory effect of PD-1. For example, when we deleted SHP2 from Jurkat cells, the ability of PD-1 to suppress IL-2 production at 6-hour time point was decreased by >50%, see Figure 1C of the revised manuscript. We observed a similar effect at even 24 hours (data not shown). These data demonstrate the importance of SHP2 for sustained PD-1 mediated inhibition, even though the PD-1:SHP2 interaction is transient. It is likely that a transient interaction at the receptor level can translate to longer term downstream effects, perhaps by modulating the SHP2 activity. Notably, similar transient interactions have also been reported for TCR and ZAP70 (PMIDs: 12356870, 27869819).

We agree that the SHP2 KO phenotype reported by Rota et al. is interesting, and suggested that Shp2 is not the sole effector for PD-1 inhibitory function in vivo. Consistent with this notion, our recent cell-based study (PMID: 32437509) also suggested the existence of SHP2-independent function of PD-1. However, while it is straightforward to assume that SHP1 mediates the SHP2-independent function of PD-1, our functional experiments using SHP1 KO and SHP1/2 double KO T cells did not support this view (see Figure 1C of the revised manuscript). These results suggest that the SHP2-independent function of PD-1 is mediated by yet-to-be identified effectors. Nevertheless, the existence of SHP2-independent functions of PD-1 does not argue against the contribution of SHP2 in the PD-1 pathway, as has been demonstrated by a number of independent studies.

I would suggest that the title be modified slightly from "SHP1/SHP2 discrimination" to "differential SHP1/SHP2 interaction" and leave discussion of discrimination until they have the functional data integrated over times that are relevant to T cell transcriptional regulation (1-2 hrs). The functional analysis can be in another paper, but it would be interesting to have a paragraph in the discussion raising the outstanding issues beyond stable binding detected by the pull-down and microcluster recruitment experiments- what are the implications for function. Could the transient interactions in the noise of the steady state and equilibrium measurements be functional?

We thank the reviewer for the suggestion. We have changed the title to “Molecular Features Underlying Differential SHP1/SHP2 Binding of Immune Checkpoint Receptors”.

With regards to the discussion whether transient interactions in the noise of the steady state can be functional, it does not seem to be the case for PD-1:SHP1 interactions, as supported by the SHP1 and SHP2 KO experiments in this study and our previous study (PMID: 32437509). From a biochemistry perspective, we feel that even SHP1 is weakly recruited by PD-1, as we observed in a subset of cells, the ‘crappy’ ITIM would not be able to efficiently release the autoinhibition of SHP1. We now add these considerations in the Discussion section of the revised manuscript (lines 397-400).

I would summarise that the work is outstanding as biochemistry and biophysics and it should be published nearly as is. I'm suggesting minor revisions in that the changes are just to text, but I think this is important and somewhat nuanced aspect of the paper that will make it even more helpful to readers.

We appreciate the positive and insightful comments! We have revised the manuscript carefully according to this reviewer’s suggestion.

Reviewer #1 (Significance (Required)):The authors generate a detailed descriptive data set about the component interaction of SHP1 and SHP2 SH2 domains with PD1 and BTLA intracellular domains. They then test hypotheses generated from the descriptive data set to better define the nature of the interactions and why PD1 recruits primarily SHP2, while BTLA mainly recruits SHP1. PD1 is a major driver or the cancer immunotherapy revolution and SHP2 is the major candidate for a signalling effector of PD1. This paper can become the reference paper for the specificity and engineering of this interaction, which will make it highly significant in a very active and still expanding field.Referee Cross-commentingI still feel that "discrimination" has a functional/activity connotation that is not addressed at all in this paper, but can be addressed. I'm happy to have the suggestion stand and let the authors decide. They need to live with it once its published. Another suggestion- the citations on regulation are mostly old. A good recent paper is Padua, R. A. P., Sun, Y., Marko, I., Pitsawong, W., Stiller, J. B., Otten, R., and Kern, D. (2018). Mechanism of activating mutations and allosteric drug inhibition of the phosphatase SHP2. Nature Communications, 9(1), 4507. https://doi.org/10.1038/s41467-018-06814-w .

We have changed the title to as the reviewer suggested. We believe that the functional questions raised by this reviewer, including the SHP1 and SHP2 contribution in PD-1 signaling, had been addressed in our current study and our recent publication (PMID: 32437509). By measuring a pembrolizumab dose response in WT, SHP1 KO, SHP2 KO, and double KO T cells, we provided evidence that PD-1 inhibitory function is contributed by SHP2, but very little if any by SHP1 (see Figure 1C of the revised manuscript).

We thank this reviewer for suggesting the excellent reference. We have cited this reference in the revised manuscript (lines 86 and 396).

Reviewer #2 (Evidence, reproducibility and clarity (Required)):In this study, Xu and co-workers investigate the biophysical nature of the interaction between the structurally-related non-transmembrane PTPs Shp1 and Shp2 with the ITIM/ITSMcontaining inhibitory receptors PD-1 and BTLA using cell-based, biochemical, biophysical and domain swapping assays. The primary aim being to better understand how these receptors discriminate between binding Shp1 and/or Shp2, and the orientation of Shp1 and Shp2 engagement. These are major unresolved questions in the field that the authors go some way to addressing in a methodical, rigorous, clear and concise manner. Findings are convincing, correlate well with previous findings and internally, and are complemented with excellent schematics, making it easy to comprehend.Major commentsThe authors focus primarily on binding affinities to explain differential binding of Shp1 and Shp2 by PD-1 and BTLA ITIMs and ITSMs, but this is only part of the story. Avidity, compartmentalization, stoichiometry of kinases, and relative abundance of Shp1 and Shp2 are also important aspects of the discriminatory mechanism that are not addressed. Competition assays would go some way to addressing the latter point and should at least be considered and discussed.

We agree that various parameters mentioned by this reviewer, such as compartmentalization and relative expression levels would be a concern for a purely cell-free system such as the SPR experiments (Figures 3 and 6), however, we feel that our cell-based assays (Figure 4B and Figures 7-9), already integrate these parameters. This is also precisely the reason why we chose to examine the recruitments of SHP1/2 in a cellular context in conjunction with cell-free systems.

Regarding the competition, we have observed generally consistent results in both WT and SHP2 KO background, with or without the potential competition from endogenous SHP2 (Figure 7B and Figure S4). Our data suggests that competition is not a dominant mechanism for the recruitment specificity we observed.

In the revised manuscript, we have clarified these points in lines 93-96.

Similarly, authors do not address how distortion of the pY binding pocket of Shp1 and Shp2 nSH2 domains in the auto-inhibited conformation is released, allowing the domain to engage with phopho-ITIM/ITSM. Again, this should at least be discussed. Current binding studies do not address this issue.

We feel that the overall recruitment to the PD-1 microclusters as we observed in cells already integrate this auto-inhibition mechanism of SHP1 and SHP2, because we used full length proteins. We do agree with the reviewer that future studies are warranted to address the contributions of each mechanism, including auto-inhibition, concentration, competition, etc., to the overall recruitment. This might require careful and extensive biophysical analyses coupled with mathematical modeling given the complex regulation of SHP1 and SHP2. As suggested by this reviewer, we have discussed the autoinhibition mechanism in multiple locations of the revised manuscript, e.g. lines 95, 395, 397.

Minor comments:Phosphorylation should be indicated in schematic representations in Figures 3, 6 b, c.

We thank the reviewer for this advice, we have now indicated phosphorylation in the revised figure 3 and 6.

Cellular and physiological significance should be further discussed, as well as broader implications of findings to other ITIM/ITSM-containing receptors in other lineages.

We agree with the reviewer that ITIM/ITSM-containing receptors have a much broader implication. As suggested, we have revised the introduction section extensively to put the work in a broader context. For example, we now mention the functions of these receptors in various cell types, including T cells, B cells, NK cells, platelets and cells of the myeloid lineage.

Reviewer #2 (Significance (Required)):Findings from this study advance our knowledge of how inhibitory checkpoint regulatory receptors discriminate between Shp1 and Shp2, which has important implications for understanding how the unique biochemical, cellular and physiological functions of these receptors and phosphatases are dictated. Indeed, findings lay the foundation for a universal mechanism, that may apply to all ITIM/ITSM receptors in other cell lineages, and perhaps novel ways of targeting these interactions therapeutically.Compare to existing published knowledgeAlthough largely correlative with previous studies, findings from this study start to fill major gaps in our knowledge of these biochemical processes, in a highly rigorous, concise and clear manner. Findings from previous studies were more 'piecemeal', whereas this study consolidates and advances important nuances of these interactions. Moreover, it lays the foundation for further structural, physiological and therapeutic studies.AudienceThe immune receptor signaling community and beyond, including any lineage in which ITIM/ITSM-containing receptors play a major role in regulating cellular responses.Your expertiseITIM/ITSM-containing receptors, kinase-phosphatase molecular switches, cellular reactivity to extracellular matrix proteinsReferee Cross-commentingGenerally agree with reviewer's comments. Constructive overall and fair. Although I was thinking additional competition experiments, I do not think necessary. Over the top for this study. Hence, 1 month should suffice to revise accordingly.

We thank this reviewer for the constructive and fair comments!

Reviewer #3 (Evidence, reproducibility and clarity (Required)):Summary:Inhibitory immune receptors containing ITIMs function through recruiting the phosphatases SHP-1 and SHP-2. SHP-1 and SHP-2 are remarkably similar yet have different roles in vivo.How can ITIM-containing immune receptors specifically recruit SHP-1 or SHP-2? In this paper, Xu et al. ask how SHP-1 vs SHP-2 specificity is achieved. They use very thorough biochemical assays to measure the affinity of SHP-1 and SHP-2 for various ITIM/ITSMs and finally pin point some key amino acids that switch an ITIM/ITSM from SHP-2 to SHP-1 specificity. The in vitro biochemical assays are augmented by in cell assays that support their conclusions. Overall, this paper is an incredibly elegant and straight forward paper addressing how SHP-1/SHP-2 specificity is achieved.Major Comments:None.Minor Comments:Could the western blots in Figure 1 be quantified as the western blots in other figures?

We have quantified the blots in Figure 1 as suggested in the revised manuscript.

The data that the y+1 reside is essential for SHP-1/2 specificity is very convincing. We are curious if the other residues of the ITIM/ITSM also contribute to this specificity, albeit less potently. The PD-1 G224A mutant is still less potent than the PD-1 BTLA ITIM swap, suggesting that while the y+1 position is most important, the other residues contribute some specificity. The authors also included data on a PD-1 variant with the BTLA ITIM A224G mutation (8f), which is slightly better at recruiting SHP-1 than the PD-1 ITIM. It may be worth mentioning this data in the text of the paper as well as displaying it in the figure.

The reviewer raised an excellent point, yes, our data does suggest that other pY-flanking residues within the ITIM also contribute to SHP1 binding. However, the pY+1 residue replacement produced the strongest effect as the reviewer noted. In the revised manuscript, we have acknowledged the potential contributions of other residues.

To further clarify this point, we performed a structural homology modeling of PD-1-pITIM:SHP1nSH2 complex based on the published crystal structure of PD-1-pITIM:SHP2-nSH2 (PDB code: 6ROY) (PMID: 32064351). The simulated structure identified a hydrophobic pocket within SHP1-nSH2 that appeared to bind nonpolar side chain at the pY+1 position. Moreover, the size of the pocket explains why a medium-sized residue would fit the best. These provide a structural interpretation for the important role of the Y+1 position. We now showed the data as Figure 9A, B in the revised manuscript.

A brief introduction to ITIM vs ITSM in the introduction of the paper may be helpful background for readers. For example, ITIM receptors are reasonably well known but how ITSM functionally differs is probably less well known.

We have rewritten the introduction about ITIM and ITSM for better clarity. Specifically, we now mention that some ITSM interacts with both SHP1/2 and SH2-containing adaptor proteins.

Although not the major focus of the paper, broadening out this SHP-1/2 specificity to other immune receptors in the discussion is fascinating. (a) The authors find that a Valine, Leucine, or Isoleucine in place of the Alanine in y+1 is very close to equivalent, yet the A is highly conserved. The authors speculate that there may be an advantage to sub-maximal SHP-1 affinity because it is more easy to regulate. I think this is reasonable speculation but a little unsatisfying given the very small observed difference in SHP-1 binding. If the authors have additional thoughts, I would be interested to hear them.

This reviewer raised an excellent point. We agree that the Valine, Leucine or Isoleucine at the pY+1 position of ITIM did not product an obvious increase in SHP1 recruitment over Alanine at the same position. To further address this concern, we have conducted new functional experiment measuring how the identity of the pY+1 residue affect the ability of PD-1 to inhibit IL-2 production from Jurkat cells. Our result showed that medium sized residues, such as Alanine, Valine, Leucine and Isoleucine, increased the PD-1 potency to a similar extent over the WT PD-1. Therefore, we have removed the questionable discussion about the “sub-maximal SHP1 affinity” in the revised manuscript. That being said, both SHP1 recruitment and IL2 suppression showed a bell shaped dependence on the side chain volume of pY+1 residue, as supported by our new structural modeling data mentioned above. We have make this point clear in the revised manuscript.

(b) The authors note that PD-1 is the only ITIM with a glycine in the Y+1 position. Are there other receptors that function primarily through SHP-2, and how might they achieve this specificity?

Among the several receptors that we tested, PD-1 is the only receptor that exhibited no recruitment of SHP1. The lack of SHP1 recruitment is also true for murine PD-1, which has an acidic glutamate residue at Y+1 position. In addition, earlier work reported that PECAM1 also selectively recruits SHP2, but not SHP1. We have noted that PECAM1 contain a threonine (polar) at the pY+1 position of their ITIMs. Thus, their inability to recruit SHP1 is consistent with our model that a nonpolar residue at Y+1 position is required for strong SHP1 recruitment. We have discussed these points in the revised manuscript.

Figure 9 b Val not Vla, Figure 3a – a legend for the color code may be nice (ie, 20-1000 nM)

We thank the reviewer for catching these, we have fixed the error in Figure 9B and provide the color code in Figure 3A in the revised manuscript.

Reviewer #3 (Significance (Required)):Significance:SHP-1 and SHP-2 play a critical role in regulating immune system function. In addition, the receptors recruiting these phosphatases (like PD-1) are important immunotherapy targets. Previously, the question of SHP-1/SHP-2 specificity has been primarily described for ITIM bearing receptors individually. Other studies have predicted consensus sequences for the tSH2 domains of SHP-1 or SHP-2, but not addressed the defining molecular characteristics of these consensus sites or how these could be combined on ITIM receptors to generate selectivity between these related phosphatases. This paper represents a significant step forward because it provides a unifying mechanism explaining how ITIM-bearing immune receptors specifically recruit SHP-1 or SHP-2. I expect this paper will be broadly interesting to biochemists, immunologists and cancer biologists.Referee Cross-commentingI generally think the other reviewers comments are reasonable and insightful. Together, they suggest no new experiments are necessary. As for the proposed title change, I prefer the authors title and find it to be justified given their data.Reviewer #4 (Evidence, reproducibility and clarity (Required)):In this manuscript, Xu and college performed an elaborate study to investigate the molecular basis of Shp1 and Shp2 discrimination by immune checkpoints PD-1 and BTLA. The paper is original, clear, and well written. I only have a few minor comments:1. Please label the molecular weights to all the western blots/IPs results.

We have labeled the molecular weights to all the blots in the revised manuscript.

2. Please add scale bars to all the microscopy pictures.

We have added scale bars to all the microscopy images in the revised manuscript.

3. For the SPR data, please add the fitting curves.

We thank the reviewer for the suggestion. However, we did not use the fitting curve to calculate the Kd, instead, we plotted the maximum response as a function of concentration to determine the Kd. This is another well accepted method for Kd calculation. In fact, some of the SPR curves fit poorly with the existing algorithm. Thus, showing the fitting curves might distract the readers.

Reviewer #4 (Significance (Required)):The strength of this paper relies on the details they dissected by using a series of mutagenesis screening experiments, which should be interesting to cell biologists and cancer immunologists.Referee Cross-commentingI think the other reviewer's comments are insightful and constructive, the suggested experiments are necessary and will improve the paper.

We thank this reviewer for the positive comments!